# Molecular quantitative trait loci in reproductive tissues impact male fertility in cattle

Xena Marie Mapel[1,4], Naveen Kumar Kadri [1,4], Alexander S. Leonard[1], Qiongyu He[1], Audald Lloret-Villas [1], Meenu Bhati[1,2], Maya Hiltpold[1,3] & Hubert Pausch [1] ✉

Breeding bulls are well suited to investigate inherited variation in male fertility because they are genotyped and their reproductive success is monitored through semen analyses and thousands of artificial inseminations. However, functional data from relevant tissues are lacking in cattle, which prevents fine-mapping fertility-associated genomic regions. Here, we characterize gene expression and splicing variation in testis, epididymis, and vas deferens transcriptomes of 118 mature bulls and conduct association tests between 414,667 molecular phenotypes and 21,501,032 genome-wide variants to identify 41,156 regulatory loci. We show broad consensus in tissue-specific and tissue-enriched gene expression between the three bovine tissues and their human and murine counterparts. Expression- and splicing-mediating variants are more than three times as frequent in testis than epididymis and vas deferens, highlighting the transcriptional complexity of testis. Finally, we identify genes (*WDR19*, *SPATA16*, *KCTD19*, *ZDHHC1*) and molecular phenotypes that are associated with quantitative variation in male fertility through transcriptome-wide association and colocalization analyses.

Male fertility varies considerably between individuals, but it remains difficult to identify underlying genetic mechanisms[1,2]. Exploring the genetic basis of reproductive success requires separating genetic from environmental effects and differentiating between male and female factors that contribute to fertilization; this is possible when repeated phenotypic measurements are available for large cohorts. While such cohorts are unavailable for nearly all species, including humans, the beef and dairy industries' reliance on artificial insemination has generated comprehensive catalogs of semen quality and male fertility records for thousands of genotyped bulls (hereafter called artificial insemination bulls) from which semen is collected at specialized artificial insemination centers[3]. Indeed, studies investigating bull fertility have high translational value as they can elucidate evolutionarily conserved functional mechanisms that contribute to mammalian

fertilization[4–6], making cattle an unconventional—but ideally suited—species to study the genetic architecture of male reproductive ability.

Bovine male reproductive traits have low to moderate heritability, suggesting they are amenable to association testing with dense molecular markers[7–10]. Datasets from artificial insemination bulls provide enough statistical power to identify moderate to large effect variants for male fertility through genome-wide association studies (GWAS). Several thousand quantitative trait loci (QTL) for male reproductive traits are listed in the AnimalQTL Database (https://www.animalgenome.org/cgi-bin/QTLdb/index); many of these loci are in non-coding regions of the genome and segregate within populations at relatively high frequencies[11–14]. Though numerous male fertility QTL have been identified in cattle, a lack of functional data has caused the molecular underpinnings for most of these loci to remain elusive[14–16].

[1]Animal Genomics, ETH Zurich, Universitatstrasse 2, 8092 Zurich, Switzerland. [2]Present address: Roslin Institute, The University of Edinburgh, Easter Bush Campus, Midlothian EH25 9RG, UK. [3]Present address: GenPhySE, Université de Toulouse, INRAE, ENVT, 31326 Castanet Tolosan, France. [4]These authors contributed equally: Xena Marie Mapel, Naveen Kumar Kadri. ✉e-mail: hubert.pausch@usys.ethz.ch

Variants that regulate gene expression and splicing (hereafter called molecular QTL, or molQTL) influence complex traits and diseases[17–20] and contribute to a significant proportion of trait heritability[21]. Several Genotype-Tissue Expression (GTEx) projects compiled comprehensive transcriptomic resources to identify thousands of molQTL throughout mammalian genomes that impact gene expression and splicing in dozens of tissues[22–24]. Integration of molQTL and phenotypic records, or summary statistics from GWAS, through transcriptome-wide association studies can identify regulatory variants that are associated with complex trait variation[25–27]. However, since this requires large uniform cohorts with detailed trait records and transcriptomics data for functionally relevant tissues, transcriptome-wide association studies for male fertility have not been feasible thus far.

Here, we establish a large and homogeneous cohort of bulls that contains both DNA sequencing and transcriptome data from testis, epididymis, and vas deferens. We demonstrate that establishing such a homogeneous cohort benefits transcriptomic profiling and identify thousands of loci that influence gene expression and splicing in three male reproductive tissues. We compare the transcriptional complexity of the three tissues and discover an excess of regulatory small-effect variants in testis that are often tissue specific. Lastly, we conduct transcriptome-wide association studies and colocalization analyses to pinpoint molecular phenotypes that impact bovine male fertility.

## Results

We sampled testis, epididymis (caput), and vas deferens tissue from 118 *Bos taurus taurus* bulls of Braunvieh ancestry (i.e., Brown Swiss, Original Braunvieh, or a cross between one of these and a different breed; Fig. 1; Supplementary Fig. 1; Supplementary Data 1). All bulls were considered post-pubertal, although the age at sampling varied from 10 to 37 months. Whole-genome sequencing of DNA extracted from testis tissue yielded between 65,207,590 and 461,834,248 cleaned paired-end reads per sample (mean = $208,761,950 \pm 63,260,972$). The subsequent alignment of these reads against the current *Bos taurus* reference genome (ARS-UCD1.2) achieved an average coverage of $12.34 \pm 3.71$-fold. Variant calling identified 29,660,795 polymorphic sites (single nucleotide polymorphisms (SNPs), insertions and deletions smaller than 50 bp), of which 21,501,032 met our quality criteria for downstream analyses (Supplementary Table 1).

Total RNA from testis, epididymis, and vas deferens tissue was extracted from the same bulls and deeply sequenced with stranded paired-end libraries. We used a splice-aware alignment tool to map an average of 257,050,199, 283,746,666, and 262,089,072 reads per sample for testis, epididymis, and vas deferens, respectively, to the bovine reference genome and the Ensembl gene annotation (Supplementary Table 2). An additional set of alignments that included WASP filtering[28] to mitigate reference allele bias contained an average of 240,544,139 mapped reads in testis, 267,645,035 mapped reads in epididymis, and 251,031,216 mapped reads in vas deferens (Supplementary Table 2). Our final dataset contained 117, 103, and 84 samples for testis, epididymis, and vas deferens, respectively, 74 of which had data of sufficient quality for all three tissues (Fig. 1; Supplementary Data 1).

### Gene expression and splicing variation is pervasive in male reproductive tissues

We estimated gene expression in transcripts per million (TPM) and, after filtering, detected 21,844 expressed genes (≥0.1 TPM and ≥6 reads in ≥20% of samples) across the three tissues. When considering the three tissues individually, we identified slightly more expressed genes in epididymis (20,376 genes; 19,561 autosomal genes) than in testis (20,222 genes; 19,440 autosomal genes) and vas deferens (19,051 genes; 18,328 autosomal genes; Table 1). A majority of expressed genes

were found in all three tissues (17,737 genes; Fig. 1E) and were protein coding (87.4%–89.7%; Supplementary Table 3). Testis had the most tissue-specific expressed genes (1106 genes), followed by epididymis (443 genes), and vas deferens (227 genes; Fig. 1E).

We examined the tissue specificity of expressed genes following an approach proposed by Djureinovic et al.[29] which considers genes that are at least 50-fold higher expressed in one tissue compared to other tissues as 'highly tissue-enriched' (Supplementary Data 2). Genes with critical roles in sperm maturation and the epididymal innate immune response against bacteria (e.g., *DEFB110*, *DEFB124*, *DEFB121*, *DEFB114*, *WFDC8*, *RNASE10*, *ADAM7*, *ADAM28*, *LCN8*, *LCN10*) were among the 50 most abundant tissue-specific (i.e., not expressed in testis and vas deferens) or highly tissue-enriched transcripts in epididymis. A gene ontology term enrichment analysis with the Protein ANalysis THrough Evolutionary Relationships (PANTHER[30]) classification system (version 17.0, http://geneontology.org/) revealed that this gene set was 20.57-fold enriched (p = 4.97e-12) for genes related to the gene ontology term "defense response to bacterium". The same gene set enrichment analysis on the 50 most abundant genes that were either specific or enriched in testis (e.g., *INSL3*, *FATE1*, *DEFB123*, *DAZL*, *ARHGAP36*) and vas deferens (e.g., *CD52*, *TNC*, *IGFBP5*, *MMP7*, *HOXD10*, *HOXD11*) did not reveal significant gene ontology terms for either tissue.

The cattle GTEx consortium[24] analyzed 60 testis transcriptomes. However, most of them lacked metadata, were collected from pre-pubertal animals, or were from individuals of non-taurine ancestry, precluding an immediate comparison with those from our homogeneous cohort. The abundance of 20,221 testis-expressed genes shared between ten cattle GTEx testis transcriptomes from post-pubertal (1.5–9 years) *Bos taurus taurus* bulls and our dataset was highly correlated (Pearson's r = 0.84). The correlation was higher (Pearson's r = 0.90) for 20,002 genes that were expressed with less than 500 TPM in our cohort. The expression of some genes differed between our cohort and the samples included in cattle GTEx (Supplementary Fig. 2), possibly due to differences in RNA sequencing strategy, sample origin, collection and storage, or variable RNA quality between each dataset. The cattle GTEx dataset contained only one vas deferens and two epididymis transcriptomes from adult bulls of taurine ancestry, preventing a meaningful comparison with our data.

We compared gene expression in bovine testis, epididymis, and vas deferens with male reproductive tract-specific gene expression in humans and mice[31]. Using Ensembl BioMART (http://www.ensembl.org/biomart/martview), we retrieved bovine orthologs for 380 and 329 human and mice reproductive-tract specific genes, respectively, of which the overwhelming majority (92.1% for human; 97.9% for mouse) were also expressed in the three bovine reproductive tissues. A large fraction of the commonly expressed genes were expressed in all three bovine tissues, but testis contributed the most and vas deferens the least number of genes to reproductive tract-specific expression (Supplementary Fig. 3). We observed several differences in gene expression between the species; for instance, *NANOS2*, which is moderately expressed in human testis (13.84 TPM), was barely expressed in bovine reproductive tissue and thus filtered from our dataset. We examined the expression pattern of the bovine orthologs for 220 genes that are testis-specific expressed in humans[29]. The overwhelming majority (218/220) of human testis-specific genes were expressed in bovine reproductive tissue and largely showed testis-biased expression (213/218), although, most were also expressed in bovine epididymis (n = 22), vas deferens (n = 2), or both other tissues (n = 192). We again identified noteworthy differences between the species. Bovine orthologs of two highly expressed testis-specific genes in humans, *DEFB119* encoding defensin beta 119 and *EPPIN* encoding epididymal peptidase inhibitor, were also highly expressed in bovine testis (median TPM *DEFB119* (ENSBTAG00000003364): 64.5; median TPM *EPPIN* (ENSBTAG00000001495): 498.2); however,

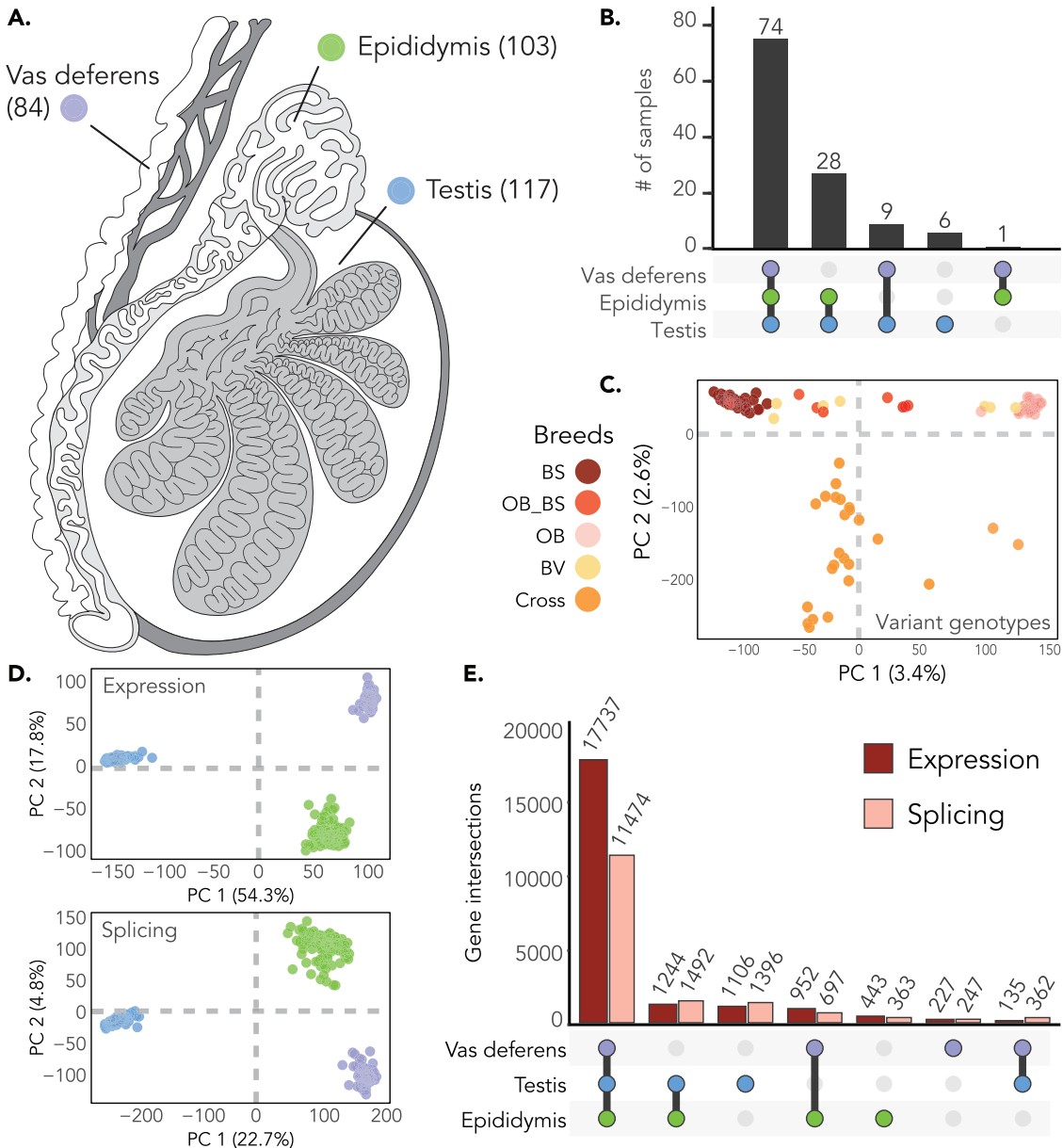

**Fig. 1 | Overview of the molQTL cohort for three male reproductive tissues.**
**A** The three male reproductive tissues used in our study and the number of samples
considered for each tissue. **B** Sample overlap across tissue types. Bar height
represents the number of individuals. Tissues are colored as in (**A**). **C** PCA of
sequence variant genotypes of 366,090 uncorrelated variants, with colors corre-
sponding to the breed assigned by the Swiss Braunvieh herd book or Cross. BV =

Braunvieh; BS = Brown Swiss; OB = Original BV; BS_OB = Cross between BS and OB;
Cross = Cross between OB or BS and another breed. **D** Scatter plot of the top two
principal components of PCAs of normalized expression values (TPM; top panel)
and normalized splicing phenotypes (PSI; bottom panel) for all tissues. Tissues are
colored as in **A**). **E** Overlap of expressed and spliced genes across the three tissues
(based off the TPM and PSI matrices).

both genes were three-fold higher expressed in bovine epididymis
(median TPM: 217.4 and 1524.3). We detected low and moderate
expression of *RNASE11* encoding ribonuclease A family member 11 in
epididymis (median TPM: 2.18) and vas deferens (median TPM: 19.06)
but did not detect any expression in bovine testis tissue despite
reports that it is highly tissue-enriched in human testis[29].

We used LeafCutter[32] to detect RNA splicing variation and calcu-
lated percent spliced-in (PSI) values for excised-intron clusters. Across
the three tissues we observed splicing variation in 16,031 genes, with
95% of these being protein-coding genes (15,219 genes; Supplementary
Table 3). We detected 11,474 genes that had splicing variation in all
three tissues. Genes that had splicing variation in all three tissues were
higher expressed than those for which we did not detect alternative
splice site usage (Supplementary Table 4). We identified 142,581 splice

junctions and 47,715 intron clusters in testis, 117,745 splice junctions
and 46,166 intron clusters in epididymis, and 107,229 splice junctions
and 41,706 intron clusters in vas deferens. The number of variably
spliced genes was similar across the three tissues, with 14,724, 14,026,
and 12,780 genes with alternative splice site usage for testis, epididy-
mis, and vas deferens, respectively. Testis has 14,243 spliced auto-
somal genes, while epididymis had 13,558 and vas deferens had 12,332
(Table 1).

The similarity of the transcriptional profile of the three male
reproductive tissues was assessed through cluster analyses. Principal
component analysis (PCA) of normalized gene expression and splicing
phenotypes separated samples by tissue type (Fig. 1C; D), which was
further supported by hierarchical clustering (Supplementary Fig. 4). In
both expression and splicing PCAs, the first principal component

**Table 1 | Results for e/sQTL mapping across tissues with a MAF filter of ≥ 1% and FDR of 5%**

|  | Testis | Epididymis | Vas deferens |
|---|---|---|---|
| Samples | 117 | 103 | 84 |
| Expressed genes | 20,222 | 20,376 | 19,063 |
| Autosomal expressed genes | 19,440 | 19,561 | 18,328 |
| eQTL | 15,642 | 4768 | 4211 |
| eGenes | 11,164 | 4347 | 3889 |
| eGenes with > 1 eQTL | 3647 | 402 | 310 |
| eVariants | 3,687,265 | 925,916 | 673,696 |
| Spliced genes | 14,724 | 14,026 | 12,780 |
| Autosomal spliced genes | 14,243 | 13,558 | 12,332 |
| sQTL | 11,450 | 3165 | 1920 |
| sGenes | 7000 | 2662 | 1718 |
| sGenes with > 1 sQTL | 2849 | 425 | 182 |
| sVariants | 2,901,402 | 683,747 | 313,771 |

separated testis from epididymis and vas deferens, suggesting that the transcriptional profile of testis is distinct from other male reproductive tissues. To further explore the similarity between tissues, we inferred pairwise similarity of gene expression and splice-junction counts with Spearman correlation. For both expression and splicing, vas deferens and epididymis were more similar to each other than to testis (epididymis x vas deferens − expression: $\rho = 0.92$, splicing: $\rho = 0.83$; epididymis x testis − expression: $\rho = 0.82$, splicing: $\rho = 0.67$; vas deferens x testis − expression: $\rho = 0.74$, splicing: $\rho = 0.57$).

**Thousands of variants impact gene expression and splicing**

To identify autosomal loci that influence gene expression (expression QTL, hereafter referred to as eQTL), normalized and standardized TPM values were regressed on variants that had minor allele frequency greater than 1.0% and were within 1 Mb of the annotated transcription start site (TSS). We accounted for covariates such as age, RNA integrity, and hidden confounders that were estimated directly from the data (PEER factors; Supplementary Fig. 5). Our analyses focused on autosomal eQTL, as the high repeat content and immature assembly of the sex chromosomes impairs the accuracy of short read mapping and variant calling[33]. We identified eQTL in at least one of the three tissues for 12,950 genes (hereafter referred to as eGenes) at a false discovery rate (FDR) of 5%. A majority of eGenes were protein-coding (11,951 eGenes; Supplementary Table 3). Testis had considerably more eGenes than epididymis and vas deferens (Table 1)−over 55% of autosomal genes expressed in testis were also eGenes. Epididymis had 4347 eGenes (22% of autosomal genes expressed in epididymis) and vas deferens had 3,889 eGenes (21% of autosomal genes expressed in vas deferens). The larger sample size for testis provided higher statistical power to identify eGenes[34,35]. However, in a subset of 74 individuals with expression data in all three tissues, testis still had 1.91 and 3.26 times more eGenes than epididymis and vas deferens, respectively; corroborating that transcriptional regulation in bovine testis is more complex than in the other male reproductive tissues. One-third of testis eGenes had more than one independent-acting eQTL (Fig. 2A), corresponding to 15,642 eQTL total and 3,687,265 unique significant variants (eVariants). We identified 4768 independent eQTL (9.2% eGenes having >1 eQTL; Fig. 2A) and 925,916 eVariants in epididymis. Vas deferens had the fewest eQTL, with 4211 independent eQTL and 673,696 eVariants. For all tissues, approximately half of all eQTL were within 100 kb of the TSS (8000 testis eQTL; 2802 epididymis eQTL; 2423 vas deferens eQTL; Supplementary Fig. 6).

The effect of each eQTL was quantified as the allelic fold change (slope_aFC) in expression of the associated eGene. The slope_aFC value was highly correlated (Spearman's $\rho \geq 0.84$) with the beta coefficient from the linear model used for eQTL mapping (Fig. 2B). Across the three tissues, only 2321 eQTL (9.4%) had large effects (defined as |slope_aFC ≥ 1|), with 65% of these loci residing within 100 kb of the eGene's TSS. Notably, testis had the smallest proportion of large effect eQTL (7.58%; 1186 eQTL; Supplementary Fig. 6). The proportion of large effect eQTL in both epididymis and vas deferens was approximately 12% (599 and 536 eQTL in epididymis and vas deferens, respectively). Large effect eQTL were enriched for low-frequency alleles (Supplementary Fig. 7). We observed that eQTL annotated as non-coding transcript variants (specifically, transcript exon variants and non-coding transcript variants), which impact non-coding RNA, had a stronger effect on expression (median slope_aFC = 0.93; Fig. 2D).

eQTL are expected to cause an imbalanced expression of the associated gene if they are cis-acting[36,37]. In this scenario, the expression of the associated eGene is influenced only by the linked eQTL allele leading to unbalanced expression of the eGene. Thus, to assess and independently quantify putative cis-regulatory effects of the eQTL, we investigated allelic imbalance in the expression of associated eGenes. In the three tissues between 78 and 80% of the variant-gene pairs were available to this evaluation, of which approximately a quarter (23−28%) showed significant evidence of allelic imbalance (FDR < 5%; Wilcoxon rank sum test). The proportion of eGenes that showed allelic imbalance was nearly twice as high (49.8, 46.8, and 47.2% in testis, epididymis, and vas deferens, respectively) for large effect eQTL (|slope_aFC| ≥ 1; $n = 1156, 669, 545$ in testis, epididymis, and vas deferens, respectively (Supplementary Fig. 8)). Approximately three quarters (76.53, 76.53, and 74.49% for testis, epididymis, and vas deferens, respectively) of the eGenes with significant allelic imbalance were within 100 kb of the eQTL (Fig. 2C). This is between 1.26- and 1.44-fold enrichment ($p < 2.16e-16$; Fisher's exact test) compared to random expectation (Supplementary Fig. 8A). The eQTL effect estimated as the magnitude of allelic imbalance was strongly correlated with slope_aFC (Spearman's $\rho = 0.69, 0.73, 0.75$ in testis, epididymis, and vas deferens, respectively). The correlation was even stronger (Spearman's $\rho = 0.87, 0.88$ and $0.88$ for testis, epididymis and vas deferens, respectively) for a subset of the eQTL where the associated eGenes showed significant allelic imbalance (Supplementary Fig. 8C).

To identify variants that influence splicing, we conducted splicing QTL (sQTL) mapping between variants within 1 Mb of the intron cluster start site and intron-excision ratios of inferred intron clusters, which were normalized and standardized within and across samples, respectively. We identified 7777 spliced genes across the three tissues with at least one sQTL (sGenes; Table 1) at an FDR of 5%−a majority of which were protein-coding genes (7471 protein-coding genes; 96% of sGenes). Testis had the most sQTL of the three tissues, further corroborating its transcriptional complexity. We detected 7000 sGenes in testis, which accounted for nearly half of its variably spliced autosomal genes. For epididymis and vas deferens, we identified 2662 and 1718 sGenes, respectively, corresponding to 20% and 14% of their autosomal spliced genes. In testis, 40.7% of sGenes had more than one independent-acting sQTL, resulting in 11,450 independent sQTL and 2,901,402 unique significant variants (sVariants; Fig. 2A). Fewer sGenes had multiple independent-acting sQTL in epididymis and vas deferens (approximately 16% in epididymis and 11% in vas deferens; Fig. 2A); in total, epididymis contained 3165 sQTL and 683,747 sVariants while vas deferens contained 1920 sQTL and 313,771 sVariants. In all tissues, between 60 and 70% of sQTL were located within 100 kb of the intron cluster start site, corresponding to 6822, 2190, and 1339 sQTL for testis, epididymis, and vas deferens, respectively (Fig. 2E).

sQTL effects were quantified as the regression coefficient from the linear model used for sQTL mapping. There were few large effect sQTL (3793 sQTL with |β-coefficient| ≥ 1) across the three tissues. Most large effect sQTL were within 100 kb of the intron cluster start site (65%, or 2476 sQTL across all tissues; Fig. 2E). The proportion of large

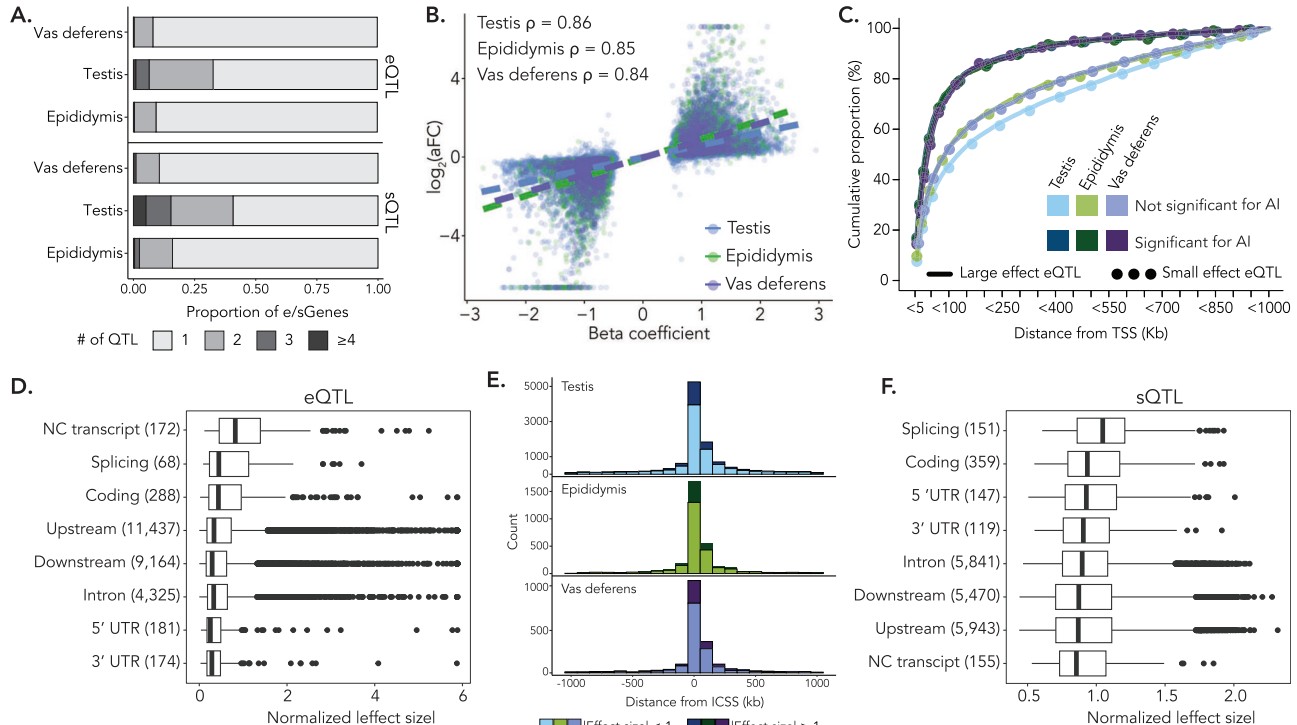

**Fig. 2 | Properties of eQTL and sQTL in three male reproductive tissues.**
**A** Proportion of independent-acting eQTL (top) and sQTL (bottom) across tissues.
**B** Correlation (Spearman's ρ) between the β-coefficient of the linear model (used for eQTL identification) and slope_aFC of each eQTL's top eVariant. The linear relationship between the two variables is shown with dotted lines. Colors correspond to the tissue type. **C** Cumulative proportion (%) of eQTL showing significant allelic imbalance in the expression of the associated eGene grouped (in bins of 5 Kb) by their proximity to the transcription start site (TSS). Data for large ($|$slope_aFC$| \geq 1$) and small ($|$slope_aFC$| < 1$) effect eQTL are presented separately. Similar data for eQTL not showing significant allelic imbalance are shown for comparison. **D** Annotation class of eQTL and rank-normalized absolute effect size for eQTL identified in all three tissues. The number of eQTL within an annotation class is listed in parentheses. **E** Distance of sQTL (top independent-acting sVariants) from the intron cluster start site (ICSS) for all three tissues. Darker colors represent large effect sQTL ($|$β-coefficient $\geq 1|$). **F** Annotation class of sQTL and rank-normalized absolute effect size for sQTL identified in all three tissues. The number of sQTL within an annotation class is listed in parentheses. The box plots in (**D**)) and (**F**)) cover the interquartile range with the median line denoted at the center, and the whiskers extend to the most extreme data point that is no more than 1.5× IQR from the edge of the box.

effect sQTL was similar across the three tissues, corresponding to 22.7%, 22.5%, and 24.9% of sQTL in testis, epididymis, and vas deferens, respectively (2603 sQTL in testis, 712 sQTL in epididymis, and 478 sQTL in vas deferens; Fig. 2E). We observed that top sVariants within splicing regions—such as splice donor and splice acceptor sites—had the largest effect sizes (median effect size = 1.14; Fig. 2F), corroborating that variants overlapping these nucleotides have pronounced impacts on pre-mRNA splicing.

Identifying distant—or trans—eQTL and sQTL that lie >5 Mb from a gene's TSS requires larger sample sizes, as they typically have small effects and an increased multiple-testing burden. Thus, we only conducted trans eQTL and sQTL analyses for testis, for which we had the largest number of samples. We observed 76 trans eGenes and 6038 trans eVariants (FDR 5%) in testis. Most trans eQTL were located on a different chromosome than the target trans eGene (73 trans eGenes and 6024 trans eVariants; Supplementary Fig. 9A). Nearly half (46%; 35 total trans eQTL) of trans eQTL were colocalized (PP4 > 0.8) with at least one cis eQTL, suggesting a complex network of cis mediation. We identified 33 intron clusters with trans sQTL, corresponding to 25 trans sGenes and 1023 trans sVariants (Supplementary Fig. 9B).

**Testis molQTL effects are unique and small, while molQTL effects in other male reproductive tissues are shared**
Tissues with similar biological functions often share molQTL[23,38]. Thus, we sought to determine if eQTL and sQTL in male reproductive tissues were shared or tissue-specific. We used Mashr to compare molQTL effects for genes that were expressed in all three tissues and

considered 11,311 and 5012 top eVariants and sVariants, respectively. For eQTL, we identified 10,738 variant-gene pairs that were significant in at least one tissue (local false sign rate <0.05). A large proportion of eQTL were tissue-specific (4760 eQTL; 44% of the eQTL considered), while there was a similar number of eQTL identified in either two or three tissues (two tissues: 2949 eQTL, 27% of eQTL; three tissues: 3029 eQTL, 28% of eQTL; Fig. 3A). Testis had the most tissue-specific eQTL, followed by vas deferens, and epididymis (Fig. 3B). eQTL effects were most similar between vas deferens and epididymis (Spearman's ρ = 0.58), and clearly less similar between testis and epididymis (Spearman's ρ = 0.32) and between testis and vas deferens (Spearman's ρ = 0.28). Conversely, most sQTL were ubiquitous—73% of the 4737 variant-gene pairs were significant (local false sign rate <0.05) in all tissues (3451 sQTL; Fig. 3A). Only 648 sQTL were tissue-specific (14% of sQTL considered; Fig. 3A), most of which were in testis (505 sQTL; Fig. 3B). We observed high pairwise similarity of sQTL effects across the three tissues, particularly between epididymis and vas deferens (testis x epididymis: Spearman's ρ = 0.75; testis x vas deferens: Spearman's ρ = 0.69; epididymis x vas deferens: Spearman's ρ = 0.88).

Tissue-specific molQTL typically have larger effects than those identified in many, or all, tissues[22,38]. We confirmed this phenomenon for epididymis- and vas deferens-specific eQTL, which had larger effects (slope_aFC) than eQTL found in all three tissues (vas deferens: Wilcoxon rank-sum $p = 2.48e-16$; epididymis: Wilcoxon rank-sum $p = 0.003$; Fig. 3C). Vas deferens-specific sQTL also had larger effects than ubiquitous sQTL (Wilcoxon rank-sum $p = 7.92e-11$), though there was no statistical difference in sQTL effect size across tissue specificity

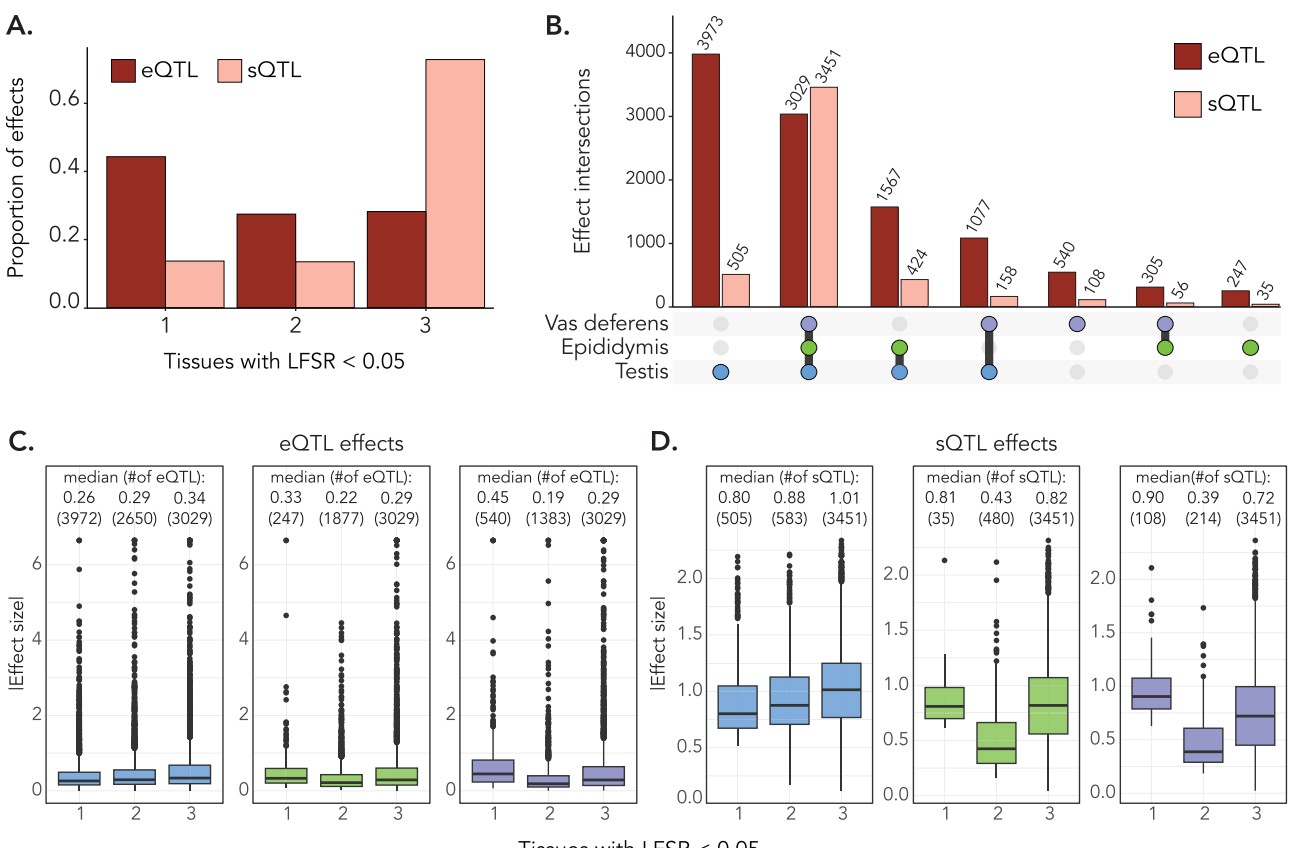

**Fig. 3 | Tissue-specific and shared effects of eQTL and sQTL. A** Proportion of eQTL and sQTL that are significant (local false sign rate (LFSR) < 0.05) in one, two, or three tissues. **B** Overlap of significant (LFSR < 0.05) eQTL and sQTL across the three tissues. Bar height represents the number of QTL. **C** Boxplots of the eQTL effect size across tissue specificity classes (significant in one, two, or all three tissues at a threshold of LFSR < 0.05) for testis (blue), epididymis (green), and vas deferens (purple) tissues. Absolute effect size corresponds to |aFC_slope|. Median effect size of each category is included. **D** Boxplots demonstrating the effect size of sQTL ($\beta$-coefficient of the linear model) for testis (blue), epididymis (green), and vas deferens (purple) tissues. Median effect size of each category is included. The number of QTL within each category is listed within brackets. The box plots in (**C**) and (**D**) cover the interquartile range with the median line denoted at the center, and the whiskers extend to the most extreme data point that is no more than 1.5 × IQR from the edge of the box.

classes for epididymis (Wilcoxon rank-sum p = 0.39). Alternatively, testis specific eQTL and sQTL had smaller effects than those shared across all three tissues (eQTL: Wilcoxon rank-sum $p = 3.93e{-}30$; sQTL: Wilcoxon rank-sum $p = 1.40e{-}16$; Fig. 3D), suggesting that testis is composed of many tissue-specific small effect molQTL.

### Integration of molecular phenotypes, molQTL, and GWAS summary statistics reveals genes associated with male fertility

We sought to investigate the impact of molQTL on male reproductive ability. First, we conducted GWAS for male fertility with a cohort of 3736 bulls that had imputed genotypes for 14,587,859 autosomal SNPs, and separately tested additive and non-additive inheritance. The fertility of the bulls was quantified through the proportion of successful artificial inseminations and accounted for genetic and environmental factors (see Hiltpold et al.[16]). While the GWAS cohort had no overlap with the molQTL cohort, both predominantly consisted of Brown Swiss bulls. Non-additive association testing revealed four male fertility QTL on chromosomes 1, 6, 18, and 26 (significance threshold of 5e−08) that contained 4,890 variants (Fig. 4A). Additive GWAS did not detect any additional significant associations and failed to identify the QTL on chromosomes 1 and 18. The QTL on chromosomes 6 and 26 were detected by the additive GWAS, but with lower significance (Supplementary Fig. 10).

We implemented the MetaXcan framework, as described by Barbeira et al.[25], to establish if variation in molecular phenotypes is associated with variation in male fertility. Briefly, S-PrediXcan was used to

integrate summary statistics from the non-additive GWAS with expression and splicing phenotypes from the three reproductive tissues. Signals were subsequently filtered based on the colocalization probability between male fertility QTL and molQTL. We identified two and seven unique genes for expression and splicing, respectively, that overlapped three of the male fertility QTL (Fig. 4, Table 2). The most striking association was on chromosome 6; *WDR19* encoding WD Repeat Domain 19 was the most significantly associated gene in epididymis and vas deferens ($p = 3.17e{-}43$ and $1.04e{-}48$). An intron cluster encompassing a canonical (58373894:58374821) and a cryptic (58373885:58374821) splice junction was the top fertility-associated molecular phenotype in both tissues (Supplementary Fig. 11). A highly significant S-PrediXcan signal ($p = 6.10e{-}38$) was also obtained for this *WDR19* intron cluster in testis, however the evidence for colocalization with the GWAS peak was low (PP.H4.abf = 0.39).

The splicing MetaXcan in testis revealed an association ($p = 1.71e{-}15$) between *SPATA16* encoding spermatogenesis associated protein 16 and male fertility (Fig. 4C). *SPATA16* mRNA was highly expressed in testis (270.57 ± 32.26 TPM), lowly expressed in epididymis (15.28 ± 9.90 TPM), and barely expressed in vas deferens (1.05 ± 1.18 TPM). Only one *SPATA16* transcript is annotated in Ensembl, but the Refseq annotation suggests three isoforms, of which the canonical isoform (XM_002684936.6) was 12 and 17 times more abundant in testis than two alternative isoforms XM_024995427.1 and XM_005201702.4 (Supplementary Fig. 12A). The fertility-associated *SPATA16* intron cluster comprised four splice junctions that span the

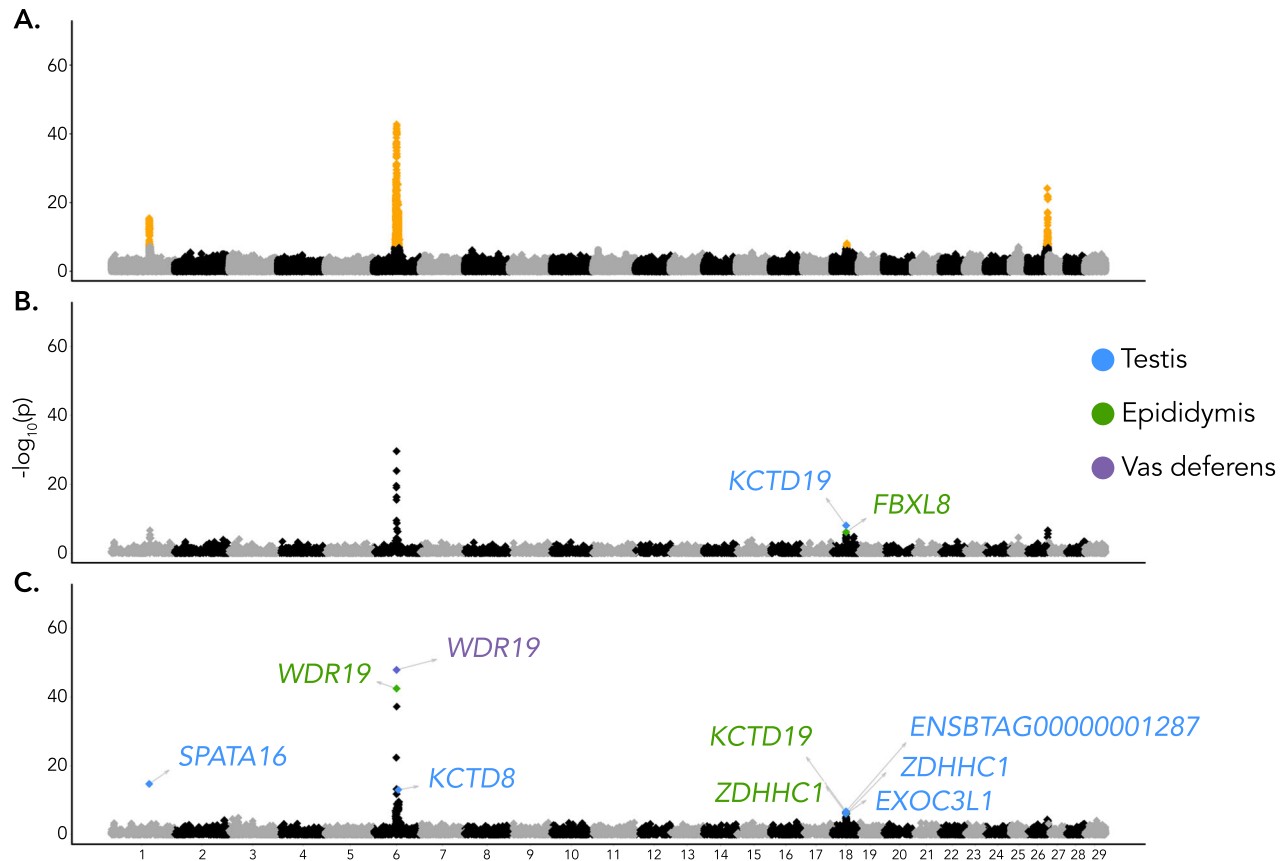

**Fig. 4 | Genes associated with male fertility.** Manhattan plots showing genome-wide association between (**A**) male fertility and imputed autosomal sequence variants in 3736 bulls from mixed linear non-additive association models. Variants that are associated with male fertility at a significance threshold of $p < 5e-8$ are highlighted in orange. Transcriptome-wide association with S-PrediXcan between (**B**) expression values or (**C**) splicing phenotypes and male fertility. Significantly associated genes after the stringent filtering based on the MetaXcan framework are marked with their names and colored according to the tissue in which associated was detected.

first four exons, including a non-coding exon that is only part of the Refseq annotation (Fig. 5C, D, F). The fertility and splicing QTL were colocalized with a probability of 0.86 and a variant (Chr1:93,954,637 bp) 365 kb upstream the annotated *SPATA16* transcription start site was the most probable causal variant (Fig. 5A). An association analysis conditional on Chr1:93,954,637 did not identify any additional variants as being associated with male fertility on chromosome 1 (Fig. 5B).

The Chr1:93,954,637 T allele compromises male fertility ($p = 4.15e-16$; Fig. 5E) but is associated with increased *SPATA16* mRNA expression ($\beta = 0.71 \pm 0.17$ TPM$_{norm}$; $p = 6.41e-05$)−albeit not at the stringent significance threshold applied to our eQTL analysis (Supplementary Fig. 12B; Supplementary Fig. 12C). Eleven of the twelve *SPATA16* exons were expressed at higher levels in animals carrying the T allele, though this difference was only significant ($p = 3.7e-06$) for the fourth exon (94,396,748 bp−94,396,893 bp). Interestingly, the second exon (94,323,018 bp−94,323,140 bp) is the only exon for which abundance is reduced ($\beta = 0.25 \pm 0.09$ TPM$_{norm}$; $p = 0.0057$) by the Chr1:93,954,637 T allele (Supplementary Fig. 12E). This exon is part of *SPATA16* isoform XM_024995427.1 but is absent in both the canonical (XM_002684936.6) and another alternative isoform (XM_005201702.4). While the Chr1:93,954,637 T allele has no effect on the abundance of the predominant canonical isoform (XM_002684936.6, $p = 0.15$), it increases ($\beta = 3.6 \pm 0.68$ TPM; $p = 6.7e-07$) abundance of XM_005201702.4 and reduces abundance of XM_024995427.1 ($\beta = -1.1 \pm 0.42$ TPM; $p = 0.015$), suggesting that it promotes differential isoform usage (Supplementary Fig. 12D). This is further supported by an inverse association between inclusion ratios of the second and third exon with the abundance of

two alternative *SPATA16* isoforms (XM_005201702.4 and XM_024995427.1; Supplementary Fig. 13).

The non-additive GWAS for male fertility revealed a QTL on chromosome 18 ($p = 6.41e-09$, lead SNP at 36,480,384) that coincides with a previously localized 3 Mb region (between 34 and 37 Mb), for which no compelling candidate causal variants were found[16]. This 3 Mb window contains 97, 27, and 12 eQTL and sQTL in testis, epididymis and vas deferens, respectively. The MetaXcan and colocalization analyses revealed five significantly associated genes (*ENSBTAG00000001287*, *EXOC3L1*, *FBXL8*, *KCTD19*, *ZDHHC1*) at this QTL, of which *KCTD19* encoding the potassium channel tetramerization domain containing protein 19 shows a highly testis-biased expression (Table 2). Two genes (*KCTD19* and *ZDHHC1* encoding the zinc finger DHHC-type containing protein 1) were each associated with two molecular phenotypes in two tissues. Three variants (Chr18:34,914,479, Chr18:34,815,920, Chr18:35,059,627), which are in moderate linkage disequilibrium (between 0.54 and 0.6), are colocalized for the male fertility QTL and seven distinct molecular phenotypes (Table 2). Association analyses conditional on any of these variants did not identify additional variants associated with male fertility on chromosome 18. The Chr18:34,914,479 T allele is associated with higher fertility, an increased *KCTD19* mRNA abundance in testis, and the expression of a *KCTD19* isoform in the epididymis that has an additional 30 bp coding sequence added to the 14th exon which results in a protein that is 10 amino acids longer than that encoded by the canonical form (Supplementary Fig. 14). *KCTD19* splicing variation was also highly significant ($p = 3.13e-15$) in testis, but MetaXcan did not reveal association of this molecular phenotype with male fertility. This

**Table 2 | Male fertility-associated genes prioritized by MetaXcan**

| Gene | Expression (TPM) | | | Splice junction and intron cluster | Molecular phenotype | Tissue | P value | PP.H4 | Colocalized variant | β-coefficient |
|---|---|---|---|---|---|---|---|---|---|---|
| | Testis | Epididymis | Vas deferens | | | | | | | |
| SPATA16 | 270.57±32.26 | 15.28±9.90 | 1.05±1.18 | 1:94345963-94396748:clu_3038 | Splicing | Testis | 1.71E-15 | 0.86 | Chr1: 93,954,637 | 0.78 |
| WDR19 | 20.09±2.65 | 29.46±19.57 | 36.26±4.82 | 6:58373894-58374821:clu_63814 | Splicing | Epididymis | 3.17E-43 | 0.96 | Chr6: 57,900,948 | -1.17 |
| | | | | 6:58373894-58374821:clu_46646 | | Vas deferens | 1.04E-48 | 0.99 | Chr6: 57,900,948 | -1.34 |
| KCTD8 | 2.71±0.73 | 0.73±1.00 | 0.68±0.93 | 6:62885148-62886027:clu_81370 | Splicing | Testis | 8.10E-14 | 0.81 | Chr6: 62,206,554 | -0.73 |
| KCTD19 | 238.01±56.20 | 9.05±6.26 | 0.58±1.42 | -- | Expression | Testis | 2.27E-07 | 0.96 | Chr18: 34,914,479 | 1.12 |
| | | | | 18:34910787-34910930:clu_25862 | Splicing | Epididymis | 2.28E-07 | 0.87 | Chr18: 34,914,479 | 0.67 |
| ZDHHC1 | 21.96±3.04 | 24.77±17.29 | 21.00±3.32 | 18:35019435-35026730:clu_33937 | Splicing | Testis | 8.11E-07 | 0.94 | Chr18:34,815,920 | -1.12 |
| | | | | 18:35019435-35026730:clu_25875 | | Epididymis | 9.44E-07 | 0.92 | Chr18: 34,914,479 | -0.81 |
| FBXL8 | 3.80±1.51 | 2.05±0.72 | 1.52±0.46 | -- | Expression | Epididymis | 7.86E-07 | 0.87 | Chr18:35,059,627 | 0.96 |
| EXOC3L1 | 13.50±2.15 | 1.34±0.55 | 1.73±0.50 | 18:34825696-34825972:clu_33908 | Splicing | Testis | 9.91E-07 | 0.93 | Chr18: 34,914,479 | 0.87 |
| ENSBTAG00000001287 | 7.17±0.97 | 3.33±1.00 | 3.26±0.81 | 18:34842264-34842709:clu_33910 | Splicing | Testis | 1.73E-07 | 0.93 | Chr18: 34,815,920 | -0.91 |

P value is the gene's p value from S-PrediXcan. PP.H4 is the probability of a shared causal variant for molQTL and QTL from colocalization analysis. Colocalized variant is the most likely colocalized variant from coloc.abf (H4.PP.SNP with largest posterior probability). β-coefficient is the effect estimate of the most likely colocalized variant from the linear model used for molQTL mapping.

variant (Chr18:34,914,479) was also associated with differential *ZDHHC1* splicing in epididymis and differential *EXOC3L1* splicing in testis.

A QTL ($p = 7.59e-23$, lead SNP at Chr26:50,145,932) for male fertility was detected on chromosome 26. Neither the splicing nor the expression MetaXcan revealed any significant gene-trait associations for this QTL.

## Discussion

We characterized the transcriptomic profile of testis, epididymis, and vas deferens in a cohort of 118 bulls. To the best of our knowledge, a cohort of this size has not been established for male reproductive tissues thus far, though smaller cohorts are available for several species[29,39–41]. Our molQTL cohort contained bulls within a narrow post-pubertal age range that predominantly belonged to one breed of European taurine cattle. We demonstrated the benefit of establishing such a homogeneous cohort for transcriptome analyses, as we detected 14-times (11,164 vs. 809) more eGenes and 4-times more sGenes (7000 vs. 1573) in testis than cattle GTEx[24] despite our molQTL cohort being only twice as large (117 individuals vs. 60 individuals in cattle GTEx). This suggests that the heterogeneity of cattle GTEx, which contains individuals from taurine and indicine populations that were sampled at different ages, and for which various RNA sequencing strategies were applied, leads to substantial underestimation of the transcriptional complexity of bovine testis and likely other tissues. The number of eGenes and sGenes identified in testis in our study is similar to what was reported in a three-times larger human GTEx cohort that also contained individuals with diverse ancestries sampled at different ages[23]. Comparing our findings with similar-sized cohorts sampled at other developmental stages and the analysis of single-cell transcriptomes will reveal spatiotemporal transcriptomic changes in male reproductive tissue, such as those occurring during testicular development and puberty[42,43].

Our comprehensive set of expressed and spliced genes enabled us to thoroughly characterize and compare the transcriptional complexity of three reproductive tissues in a large mammal, which was so far not possible due to a lack of epididymis and vas deferens data in all existing GTEx cohorts[22–24]. We revealed 21,844 expressed and 16,304 spliced genes across the three tissues. Expressed and spliced genes were generally shared across the three reproductive tissues; however, the transcriptional profiles of vas deferens and epididymis were similar, while that of testis was distinct. We found good consensus between pairs of bovine-mouse and bovine-human orthologous genes with male reproductive tract-specific expression, thus demonstrating that our molQTL cohort is a valuable resource for translational research in mammalian male reproductive biology. We observed a similar pattern for eQTL and sQTL—the effects of regulatory variants in vas deferens and epididymis were similar and often shared, whereas testis had a distinct and exceptionally large array of eQTL and sQTL. When compared to the other tissues, we found between 3- and 6-times the number of eGenes and sGenes in testis, many of which were regulated by multiple eQTL and sQTL, respectively, which emphasizes its transcriptional complexity that has been reported previously[44,45]. Testis contained thousands of tissue-specific, small effect molQTL. An abundance of molQTL in testis was observed in numerous mammalian species[22,23], though the reason for this remains unclear. The accumulation of small effect molQTL in testis may be a result of male sexual selection and sperm competition, which is driven by a more permissive transcription[46], relaxed pleiotropic constraints, and other selective mechanisms[46,47]. In humans, when compared to other tissues, testis eQTL were more likely to contain signatures of positive selection[48]; furthermore, tissue specific eQTL that showed evidence of positive selection had small effect sizes[48]. Thus, functional variants in testis may experience unique selection pressures that are essential for male reproduction. Further research is

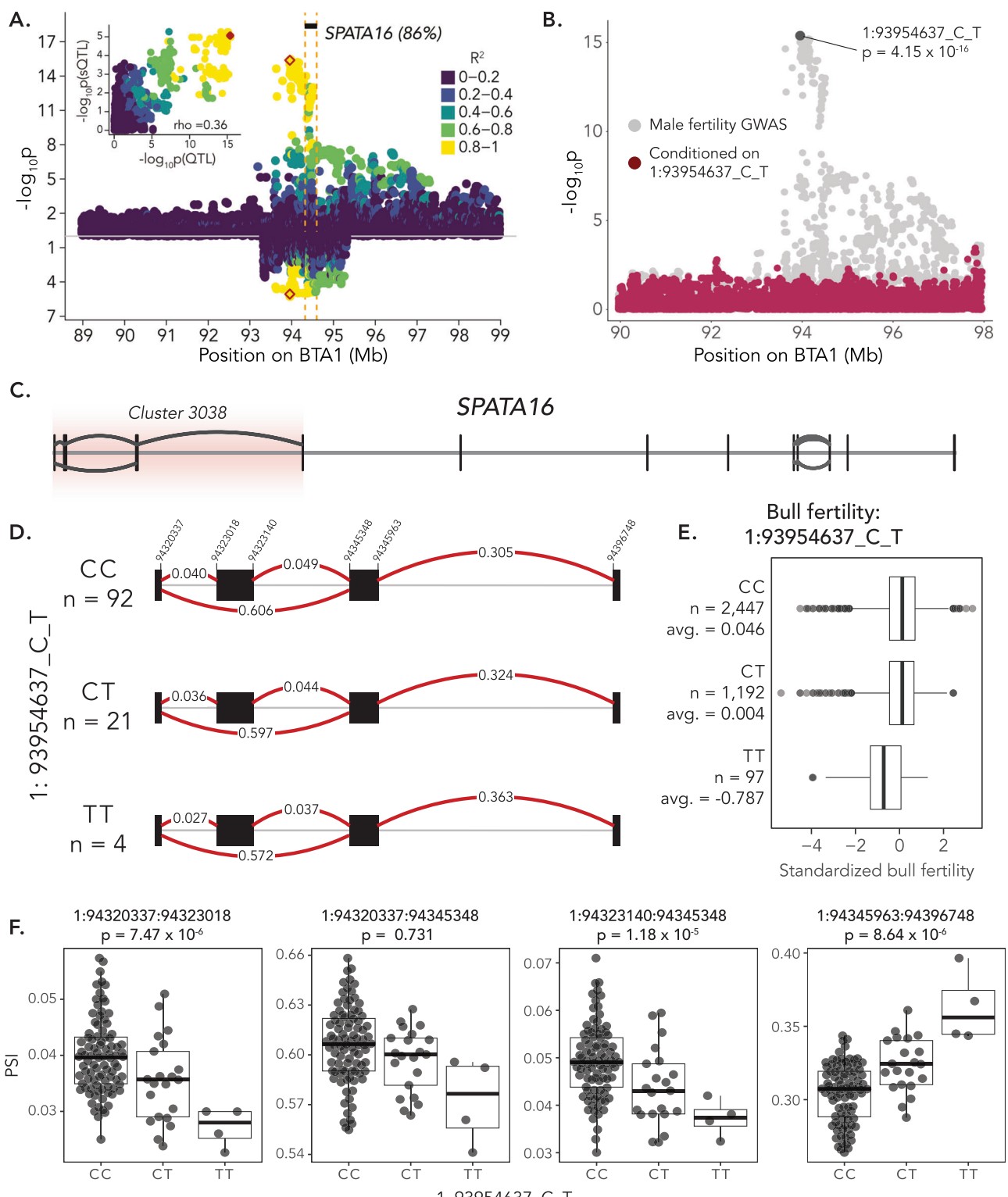

warranted to fully understand the genomic basis of complex functional patterns of the testis.

By leveraging our comprehensive array of molecular phenotypes from reproductive tissues and fertility records from genotyped artificial insemination bulls through gene-based association testing and colocalization analyses, we identified candidate causal regulatory mechanisms underpinning three of the four male fertility QTL. Among them is a previously suspected candidate causal molecular phenotype of *WDR19*[49], which adds additional confidence that this approach provides biologically meaningful associations. Our results corroborate

that gene splicing and expression variation have substantial effects on complex traits[21] and highlight that regulatory variants need to be considered thoroughly as candidate causal variants. The male fertility QTL were associated with two large effect and one small effect molQTL, thereby demonstrating that molQTL effect size may be a poor predictor of impact on complex trait variation. Spermatogenesis, sperm maturation, and sperm transport are pivotal functions of the three reproductive tissues considered. Gene-based association testing with traits such as sperm morphology or sperm motility may reveal additional gene-trait associations, as they are closer related to the

**Fig. 5 | A molecular *SPATA16* phenotype overlaps with a male fertility QTL.**
**A** Mirrored plots of −log₁₀(p)-values from association testing between imputed sequence variants and bull fertility in 3736 bulls (top, mixed linear model) and splicing phenotypes in 117 bulls for intron cluster 3038 of the gene *SPATA16* (bottom, linear regression model). The inset shows the correlation between GWAS and sQTL −log₁₀(p)-values. Color indicates the linkage disequilibrium (R²) between the most likely colocalized variant (1:93954637_C_T, red colored) and all other variants. **B** Association testing between imputed sequence variants and male fertility in 3736 bulls before (gray) and after (maroon) conditioning on the genotype of the most likely colocalized variant (1:93954637_C_T). *P* values are from a linear mixed model that included a genomic relationship matrix to control for relatedness among samples. **C** Gene structure of bovine *SPATA16*, showing positions of exons (black

lines) and splicing events (arcs). **D** Splicing within *SPATA16* intron cluster 3038 for different genotypes of the top colocalized variant. Red lines show the different splice junctions with the values representing percent spliced in (PSI) estimates within the intron cluster. **E** Standardized bull fertility across the genotypes of the colocalized variant. **F** PSI values for different splicing events within Cluster 3038 across different genotypes of the most likely colocalized variant. *P* values were from a linear model which regressed PSI values on additively coded genotypes of 117 bulls and which considered three genotype principal components, age, RIN, and ten PEER factors as covariates. The box plots in E) and F) cover the interquartile range with the median line denoted at the center, and the whiskers extend to the most extreme data point that is no more than 1.5× IQR from the edge of the box.

biological function of these tissues than male fertility. However, establishing cohorts with repeated semen quality measurements that are large enough to identify loci that explain only a small proportion of trait variation is challenging in cattle—and almost impossible in most other species.

We identified a *SPATA16* splicing event in testis that was significantly associated with male fertility. The colocalized variant fell within an intergenic region between *SPATA16* and *NLGN1*. Previously, a GWAS-only approach prioritized a missense variant of *SPATA16* as a putative causal variant[16]. However, by combining molecular phenotypes, molQTL analyses, and GWAS—which were appropriately modeled to QTL inheritance—we identified a splicing and regulatory mechanism that is significantly more likely to underpin the QTL. While loss-of-function alleles of *SPATA16* lead to severe sperm defects that prevent fertilization[50,51], our results demonstrate that variants mediating *SPATA16* expression and splicing can contribute to quantitative variation in male fertility as they are less deleterious to protein function. We also identified several gene-trait associations that showed colocalization between molQTL and a non-additive male fertility QTL on chromosome 18. Our inability to resolve this QTL may be due to high linkage disequilibrium or possibly indicates that the causal variant affects multiple genes, as may be the case for large structural variants[33,52]. However, MetaXcan prioritized genes that were previously implicated in male fertility disorders, including *KCTD19* and *ZDHHC1*[53–55]. Future fine-mapping efforts that consider additional phenotypic observations and complex genotype variations, such as structural variants called from long read cohorts, will be required to fully describe the molecular underpinnings of this QTL[33].

# Methods

## Ethics statement
Tissue of male *Bos taurus taurus* animals was sampled at a commercial abattoir. The decision to slaughter the bulls was independent from our study. None of the authors of the present study were involved in the decision to slaughter the bulls. No ethics approval was required for this study.

## Tissue sampling and DNA/RNA extraction
Testes from 128 bulls were collected at a commercial abattoir in Zürich, Switzerland after regular slaughter, then transported to a laboratory for preparation. Testis, epididymis (caput), and vas deferens tissue samples were flash-frozen in liquid nitrogen and stored at −80 °C until DNA and RNA extraction. The time from slaughter to freezing ranged from 40 min to 270 min (average: 126 min ± 47 min). Sample collection took place during the fall, winter, and spring of 2019, 2020, and 2021. Breed was assigned according to entries from the Swiss Braunvieh herdbook and the animal traffic database.

We extracted DNA and total RNA from testis tissue with Qiagen AllPrep Mini kits (Qiagen, Hilden, Germany). Frozen testis tissue was homogenized with a MagNA Lyser (Roche, Basel, Switzerland) at 6000 rpm for 50 s (x2) in RTL Plus Buffer with β-mercaptoethanol. DNA extraction followed standard manufacturer protocols.

Concentration was estimated with a Qubit 2.0 fluorometer (Thermo Fisher Scientific) prior to submission for sequencing. Extraction of testis total RNA followed manufacturer protocols. Epididymis and vas deferens total RNA was extracted from frozen tissue with Qiagen RNeasy Mini Kits (Qiagen, Hilden, Germany) and included an on-column DNase digestion with Qiagen RNase-Free DNase (Qiagen, Hilden, Germany). Frozen epididymis tissue was homogenized with the same conditions as described for testis tissue. We optimized the homogenization and lysis of vas deferens tissue by adding an additional homogenization period at 6000 rpm for 50 s and a 60-min waiting period on a cold block. We assessed the quality of testis, epididymis, and vas deferens total RNA with an RNA integrity number (RIN) inferred from a Bioanalyzer RNA 600 Nano assay (Agilent Technologies). For subsequent analyses, we only considered samples with RIN > 4.0.

## DNA and RNA sequencing
Genomic libraries (paired-end, 150 bp) for sequencing were prepared with the Illumina TruSeq DNA PCR-Free protocol and sequenced on two Illumina NovaSeq 6000 S4 flowcells. Paired-end total RNA libraries (150 bp) were prepared with the TruSeq Stranded Total RNA protocol and included rRNA depletion with Ribo-Zero Plus. Libraries were sequenced on four S4 flowcells and one S2 flowcell of the Illumina Novaseq 6000.

## DNA alignment and variant calling
We used fastp (v0.19.4)[56] with default parameters to remove adapter sequences and low-quality bases, and trim poly-G tails from raw DNA sequence data. After quality control, we aligned reads to the ARS-UCD1.2 reference genome (https://www.ncbi.nlm.nih.gov/datasets/genome/GCF_002263795.1/)[57] with the mem-algorithm from BWA (v0.717)[58] and the -M flag. We sorted the aligned reads by coordinates with Sambamba (v0.6.6)[59] then combined the read-group specific BAM files to create sample-specific sorted BAM files. We marked duplicate reads in the sample specific sorted BAM files with the MarkDuplicates module from Picard tools (v2.25.7, https://broadinstitute.github.io/picard/). We used mosdepth (v0.3.6)[60] to infer the sequencing coverage at a given genomic position, which was then used to estimate the average coverage per sample. Only high-quality reads (mapping quality >10 or without SAM flag 1796) were considered when calculating the average coverage. We called variants with DeepVariant (v1.3)[61] using the WGS mode. Called samples were merged using GLnexus (v1.4.1)[62] with the DeepVariantWGS configuration. Variants that had a missingness rate greater than 50% and variants for which the genotypes deviated from Hardy-Weinberg proportions (*P* < 1e−08) were removed. We applied Beagle (v4.1)[63] to impute sporadically missing genotypes and infer haplotypes, then removed sites with model-based imputation accuracy <0.5. To assess relationships among individuals, we constructed a genomic relationship matrix with the genotypes from 366,090 uncorrelated variants (obtained from Plink v1.9[64] with --indep-pairwise 1000 5 0.2) that had MAF > 0.5% and performed a PCA with the pca function in QTLtools (v1.3.1)[65].

## RNA alignment

We removed adapter sequences, low-quality bases, poly-A tails, and poly-G tails from the RNA sequence data with fastp[56]. We split cleaned reads into read groups with gdc-fastq-splitter (https://github.com/kmhernan/gdc-fastq-splitter) then aligned them to the cattle reference genome (ARS-UCD1.2) and the Ensembl gene annotation (release 104, https://ftp.ensembl.org/pub/release-104/gtf/bos_taurus/) with STAR (version 2.7.9a)[66]. To account for allelic mapping bias in the sQTL and allele specific expression (ASE) analyses, we produced additional alignments with the flag --waspOutputMode and heterozygous SNPs[28]. We performed coordinate sorting and added read groups with Picard tools (https://broadinstitute.github.io/picard/). We merged each sample's BAM files with Sambamba (v0.6.6)[59] and marked duplicate reads with the MarkDuplicates module from Picard tools. Statistics on the alignment quality were inferred with the flagstat function from Sambamba for each sample.

## Expression and splicing quantification

We quantified gene-level expression with the quan function in QTLtools[65]. We produced gene-level TPM values with high quality reads that passed all alignment quality filters and only included reads that were uniquely mapped and properly paired. Gene-level read counts were inferred with featureCounts (v2.0.3)[67]. We filtered expression values for each tissue to include genes with expression ≥0.1 TPM and ≥6 reads in ≥20% of samples. The filtered expressed genes were inverse normal transformed and quantile normalized for subsequent analyses[68]. To observe clustering within and across tissue types, we used the filtered, normalized TPM values to perform a PCA and hierarchical clustering with the dendextend (v1.16.0) package in R. Clustering analyses did not reveal any obvious outliers or batch effects (Supplementary Fig. 4C).

To infer splicing events, we quantified intron-excision values for identified intron clusters. We extracted exon-exon junctions from reads that passed WASP filtering with Regtools (v0.5.2)[69] and considered a minimum anchor length of 8 bp, minimum intron size of 50 bp, and maximum intron size of 500,000 bp. Intron clustering, calculation of intron excision ratios, filtering, and preparation for sQTL mapping was performed with LeafCutter (v0.2.9)[32] and executed with the cluster_prepare_fastqtl.py wrapper provided by the human GTEx consortium (https://github.com/broadinstitute/gtex-pipeline/tree/master/qtl/leafcutter). We assigned clusters to annotated genes with an altered map_clusters_to_genes.R script that accounted for strandedness and gene coordinates from a collapsed gene annotation for cattle (Ensembl release 104; generated with scripts from https://github.com/broadinstitute/gtex-pipeline/tree/master/gene_model). We limited our dataset to high-quality introns by imposing stringent filters that removed introns without read counts in >50% of samples, introns with low variability across samples, and introns with fewer than max (10, 0.1$n$) unique values (where $n$ is sample size). To normalize the filtered counts and produce files for sQTL mapping, we used the prepare_phenotype_table.py script from Leafcutter (https://github.com/davidaknowles/leafcutter). We performed hierarchical clustering and conducted a PCA on the resulting normalized intron excision phenotypes and hierarchical clustering with the dendextend package in R. Clustering analyses of splicing phenotypes did not reveal obvious batch effects or outliers (Supplementary Fig. 4D).

## eQTL mapping, allelic imbalance, and sQTL mapping

eQTL mapping was performed on the 29 bovine autosomes for all three tissues with QTLtools[65]. To account for population structure and other covariates that influence gene expression, normalized expression phenotypes were corrected with the first three genotype principal components, ten PEER[70] factors (estimated for each tissue of interest; Supplementary Fig. 5), the age of the individual, and RIN value of the sample. We considered sequence variants with minor allele frequency (MAF) > 1% and included variants 1 Mb up- and downstream of the TSS. To account for multiple testing, we used the permute function in QTLtools to conduct 1000 permutations and produce beta corrected $p$ values. We used the adjusted $p$ values and the qtltools_runFDR_cis.R script that is distributed with the QTLtools package to apply an FDR threshold of 5% and identify expressed genes with at least one significant eQTL. To identify all significant variant-gene pairs, we followed the procedure detailed in Delaneau et al.[65]. Specifically, each gene's beta $p$ values and most-likely beta parameters were used with the FDR level to feed the beta quantile function. This generated nominal $p$ value thresholds for each expressed gene, and all variants below this threshold were deemed significant. To identify genes with multiple independent acting eQTL, we implemented the conditional analysis from QTLtools. Briefly, this process uses the aforementioned nominal p-value thresholds to identify all significant eVariants, then implements forward and backward stepwise linear regression to identify independent acting eQTL. The significant eVariants are then assigned to the independent signals.

To estimate the effect size of significant eQTL, we computed the log allelic fold change (slope_aFC[71] with aFC.py (https://github.com/secastel/aFC) for the top variant of each independent eQTL. We used gene counts from featureCounts, which were normalized with DESeq2 (v1.34.0)[72] and log transformed. We compared the normalized eQTL effect sizes to the annotation category of the top variant (obtained with the Variant Effect Predictor (VEP, v109.3) from Ensembl[73]) to investigate functional consequences.

We studied allelic imbalance using RNA sequencing reads filtered with WASP[28] to mitigate reference allele bias. Whole genome DNA sequence data phased with Beagle (v5.4)[74] were complemented with the read backed phasing of the RNA seq data using phASER (v1.2.0)[75] with the flag --baseq 10. Sequencing reads overlapping heterozygous sites within protein coding genes (Ensembl version 104) were used to produce gene level haplotypic expression counts using phASER_gene_AE.py with the −min_haplo_maf 0.005 flag. A median number of 3834, 3345, and 1353 RNA sequencing reads per gene, and 6,352,369, 8,453,661 and 6,739,602 reads per sample covering the heterozygous sites of the protein coding genes were available in testis, epididymis, and vas deferens, respectively. Haplotype expression counts across all samples were compiled using phaser_expr_matrix.py to validate cis-regulatory effects of eQTL identified in the three tissues and to independently estimate their effects. eGenes harboring heterozygous sites covered with at least 8 sequencing reads and top eVariants with at least 10 homozygous and 10 heterozygous individuals were considered for this assessment, resulting in a subset of 12,261, 3832, and 3245 variant-gene pairs in testis, epididymis, and vas deferens, respectively. Gene level haplotypes were then phased with their associated top eVariant using phaser_cis_var.py. Individual level magnitude of allelic imbalance was estimated as log allelic fold change using phaser_cis_var.py. The log allelic fold change is the ratio of gene haplotype counts linked to alternative allele to the gene haplotype counts linked to the reference allele of the top eVariant on log2 scale. The median of log allelic fold change in individuals heterozygous for the top eVariant is presented as an independent estimate of the cis regulatory effect of that eQTL. Comparison of log allelic fold change in individuals heterozygous for the top eVariant allelic imbalance expected) and homozygous for the top eVariant (allelic imbalance not expected) was used to test the significance of allelic imbalance using a Wilcoxon ranksum test implemented in phaser_cis_var.py.

Autosomal sQTL mapping was performed with QTLtools[65] and generally followed the approach used for eQTL discovery. However, we used the grp-best option to adjust the permutation scheme to correct for multiple splice junctions and intron clusters within a gene. Mapping was conducted with variants 1 Mb up and downstream of the TSS and we included three genotype principal components, age, RIN, and ten PEER factors as covariates. We conducted the QTLtools

conditional analysis to identify all significant variants (sVariants) and independent acting sQTL. We considered the β–coefficient from the QTLtools linear model for the top sVariant as the sQTL effect size. Normalized sQTL effect sizes were compared across functional annotation categories using the previously described variant annotation classes.

Trans eQTL and sQTL were only mapped for testis. We considered 5,685,508 variants that had MAF > 5%, were outside of repetitive regions (identified by RepeatMasker (v4.1.4); http://www.repeatmasker.org), had imputation accuracy of 1.0, and had mappability > 1. Variant mappability was determined with GenMap (v1.3.0)[76] using kmers with lengths of 36 for UTRs and 75 for exons (based on the ARS-UCD1.2 reference) and allowed for two mismatches. We mapped trans QTL for protein coding genes that had a mappability >0.80 (18,876 genes). Gene mappability was estimated with crossmap (v1.2, https://github.com/battle-lab/crossmap)[77]. Phenotypes (either TPM values or PSI values) were corrected for the same covariates used in the cis analysis and rank normal transformed. We used the trans function from QTLtools to identify QTL > 5 Mb from a gene's TSS and performed 100 permutations, which were then ranked and used to estimate FDR. We considered a trans eVariant or sVariant significant if it had an FDR < 0.05. We removed trans QTL that were cross-mapped (https://github.com/battle-lab/crossmap)[77]. Due to excessive linkage disequilibrium observed in cattle, we removed trans QTL with variants that were in linkage disequilibrium >0.01 with eQTL or sQTL for those eGenes or sGenes, respectively.

### Shared eQTL and sQTL
We identified shared and tissue specific QTL with Mashr (v0.2.69)[38]. We included top eVariants and sVariants from genes that were expressed in all three tissues. To prevent the double-counting of molQTL that are in linkage disequilibrium and impact the same gene or intron cluster, we selected a single top molQTL that had the largest effect across tissues (as described in Urbut et al.[38]). This resulted in 11,311 eVariants and 5012 sVariants. We considered a subset of 200,000 randomly selected variants and invoked a threshold of local false sign rate <0.05 to establish if an effect was significant in a tissue. For significant effects, we assessed pairwise tissue similarity with Spearman's correlation. We compared the magnitude of molQTL effect sizes across specificity classes (tissue specific, in two tissues, or in all tissues) for each tissue with a Wilcoxon signed rank test.

### GWAS of male fertility
Genome-wide association tests assuming additive and non-additive modes of inheritance were carried out between imputed genotypes and estimates of sire insemination success (which we used as a proxy for male fertility)[16] in 3736 bulls using GCTA (v1.92.1)[78]. The additive association testing was carried out using additively coded SNP genotypes. For the non-additive association testing, SNP genotypes were coded assuming recessive inheritance to the reference (R) allele and alternate (A) allele. Genotypes were coded as RR = 1, RA = 0, AA = 0 (M1) in the former case and as AA = 1, RA = 0, RR = 0 (M2) in the latter case. Only SNPs with MAF > 0.5% and where the frequency of 1's was between 0.5% and 0.95% ($N$ = 14,587,856) were retained for association testing. The smallest $P$ value from M1 and M2 was considered as the strength of non-additive association between genotypes and phenotypes for each of the tested variant, and variants with $P$ < 5e−08 were considered significant. A genomic relationship matrix constructed from 18,187,234 autosomal SNPs with MAF > 0.005 was included in the association model to control for relatedness among samples.

### Integrating QTL and molQTL in MetaXcan framework
We used S-PrediXcan (v0.7.5)[25] to test for an association between molecular phenotypes (expression and splicing phenotypes) and male fertility. The summary statistic-based approach was applied to include results from non-additive association testing. Briefly, we trained elastic-net based prediction models for expression values (TPM) and splicing phenotypes (PSI), respectively, for all genes and intronic clusters separately in testis, epididymis, and vas deferens. SNPs located within 1 Mb of the TSS of the genes were considered for training with the script gtex_tiss_chrom_training.R. In total, 18,053,560 sequence variants (MAF > 1%) called in the molQTL cohort were available for training the prediction models. Covariates used in the eQTL and sQTL scans were included in training of respective prediction models to account for confounding factors. High-performance prediction models (zscore_pval <0.05 and rho_avg >0.1) were available to predict expression levels of 15,810 genes and to predict splicing phenotypes for 46,426 intron clusters across three tissues (77, 80, and 58% of eGenes and for 92, 94, and 80% of intron clusters in testis, epididymis, and vas deferens respectively). Using weights for each SNPs in these high-performance prediction models, we then carried out gene level and cluster level association test for expression and splicing phenotype respectively using the script SPrediXcan.py. Following the recommendations in Barbeira et al.[25], Bonferroni-corrected $P$ values of 3.16e−06 (0.05/15,810) and 1.08e−06 (0.05/46,426), respectively considering the number of genes and intron clusters tested for expression and splicing phenotypes, were set as a significant threshold. Further, a filter was applied on the prediction performance of the model with the threshold set at 0.05/number of significant results after the previous filtering.

Next, to avoid capturing linkage disequilibrium-contaminated associations, as recommended in the MetaXcan framework[25], we carried out Bayes Factor colocalization analyses with the R package coloc (v5.1.0.1)[79]. For every significant GWAS signal ($p$ < 5e−08), we searched for evidence of its colocalization with molQTL using the coloc.abf function. The search space was limited to 5 Mb up- and downstream of the significant GWAS peaks. Colocalizations with PP.H4.abf (probability of a shared causal variant) > 0.8 and PP.H3.abf (probability of different causal variants) <0.5 were retained and used to filter results from the transcriptome-wide association studies.

### Characterization of a SPATA16 sQTL
The Ensembl annotation of *SPATA16* contains only one transcript, whereas the Refseq annotation suggests three isoforms (XM_002684936.6, XM_024995427.1, XM_005201702.4) that differ in both the number of coding and the number of non-coding exons. The abundance of the three isoforms in bovine testis was quantified in this study with kallisto (v0.46.1)[80]. Testis RNA sequencing data was processed and aligned to the cattle reference genome (ARS-UCD1.2) and the Refseq gene annotation (version 106, ftp://ftp.ncbi.nlm.nih.gov/refseq/B_taurus/annotation_releases/106/GCF_002263795.1_ARS-UCD1.2/GCF_002263795.1_ARS-UCD1.2_genomic.gff.gz) with STAR (see above). Exon-specific *SPATA16* expression was quantified with QTLtools quan on the STAR-aligned bam files (see above). Association tests between exon expression and Chr1:93,954,637 were conducted with a linear model in R that considered three principal components, age, and RIN as covariates. Bonferroni correction was applied to determine a significance threshold with 0.05/n, where n is the number of exons tested.

### Statistics and reproducibility
No statistical method was used to predetermine sample size for any of the analyses. Tissue was collected from randomly sampled animals at a commercial slaughterhouse. All sequencing data that fulfilled minimum quality parameters were considered. No data were excluded from the subsequent analyses.

### Reporting summary
Further information on research design is available in the Nature Portfolio Reporting Summary linked to this article.

## Data availability

DNA and RNA sequencing data of 118 bulls are available in the ENA database at the study accessions PRJEB28191 (Short read sequencing of cattle) and PRJEB46995 (Testis transcriptome of mature bulls). Comprehensive metadata for all samples are available as Supplementary Data 1. Gene expression and splicing matrices, a VCF file of genome-wide genotypes used for e/sQTL mapping, a cross-table to link genotype and transcriptome data, results from trans-e/sQTL mapping as well as summary statistics from additive and non-additive GWAS used for transcriptome-wide association testing have been archived at zenodo (https://zenodo.org/records/10409025). Gene expression data (cGTEx_TPM_8646sample_27607gene.txt.gz) and corresponding meta data (cGTEx_meta_data_8646sample.xlsx) processed by the cattle GTEx consortium are available at zenodo (https://zenodo.org/records/7560235). Human and mice reproductive tract specific genes are available in Tables S5 and S6 from Robertson et al.[31]. Human testis specific genes are available in Supporting Table 4 from Djureinovic et al.[29]. The bovine reference sequence ARS-UCD1.2 is available at https://www.ncbi.nlm.nih.gov/datasets/genome/GCF_002263795.1/. The Ensembl gene annotation (release 104, Bos_taurus.ARS-UCD1.2.104.chr.gtf.gz) is available at https://ftp.ensembl.org/pub/release-104/gtf/bos_taurus/. The Refseq gene annotation (version 106, GCF_002263795.1_ARS-UCD1.2_genomic.gff.gz) is available at https://ftp.ncbi.nlm.nih.gov/refseq/B_taurus/annotation_releases/106/GCF_002263795.1_ARS-UCD1.2/. Raw data to reproduce the figures are either available at zenodo (https://zenodo.org/records/10409025) or provided in the Source Data file. Source data are provided with this paper.

## Code availability

All software used to process the data have been referenced in the Methods section.

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

## Acknowledgements

This study was supported by grants from the Swiss National Science Foundation (HP, grant ID: 310030_185229), an ETH Research Grant (HP), Swissgenetics (HP), and the European Union's Horizon 2020 research and innovation programme under Grant Agreement No. 815668 (Bov-Reg; HP). The funding bodies were neither involved in the design of the study and collection, analysis, and interpretation of data nor in writing the manuscript. We appreciate support from the abattoir veterinarians and staff as well as Till Graf, Patrick Fitzi, and Remo Hengartner for assistance with tissue sampling. We thank Dr. Fritz Schmitz-Hsu and Dr. Ulrich Witschi (Swissgenetics) for providing phenotype data. We thank Braunvieh Schweiz for providing genotype data of Swiss BSW bulls. We acknowledge the Arbeitsgemeinschaft Deutsches Braunvieh, Braunvieh Austria, Tierzuchtforschung Grub, the Chair of Animal Breeding of TU München, the Institute of Animal Breeding from Bayerische Land-esanstalt für Landwirtschaft and ZuchtData EDV Dienstleistungen GmbH for providing genotype and phenotype data of Austrian and German BSW bulls. We thank the Functional Genomics Center Zurich (Dr. Cath-arine Aquino) for generating DNA and RNA sequencing data.

## Author contributions

X.M.M. sampled tissue and established the molQTL cohort, aligned RNA sequencing data, quantified gene expression and splicing variation, developed workflows and applied them to map e/sQTL, contributed to the TWAS, interpreted results, and drafted the manuscript; N.K.K. imputed sequence variant genotypes, established workflows for GWAS, conducted ASE analyses and TWAS, contributed to eQTL analysis, interpreted results, and contributed to the writing of the manuscript; N.K.K. and X.M.M. performed colocalization analyses; M.H. coordinated the sampling of tissue with the abattoir staff, sampled tissue, and pre-pared phenotype data; A.L.V. contributed to short read alignment; A.S.L. performed variant discovery and genotyping, and contributed to e/sQTL mapping; QH contributed to GWAS; M.B. performed the WASP align-ment of the RNA-seq data and contributed to discussions about the molQTL mapping workflows; H.P. conceptualized the study, contributed to the molQTL fine-mapping, interpreted results, and contributed to the writing of the manuscript; all authors read and approved the final version of the manuscript.

## Competing interests

The authors declare no competing interests.
