## [Peer Review File · Nature Communications]

Molecular quantitative trait loci in reproductive tissues impact male fertility in cattleReviewer #1 (Remarks to the Author):

This study uses whole genome sequencing and transcriptomic analyses of testis, epididymis, and vas deferens tissues to perform eQTL, splice eQTL and TWAS analyses to understand the molecular drivers of fertility traits in bulls of Braunvieh ancestry.

There are very few datasets with matched whole genome sequence and transcriptomic data of this scale and quality for farmed animals and the authors emphasise the opportunity that the study provides in leveraging this type of data for complex trait prediction, particularly for a trait which is mid- to -lowly heritably such as fertility.

The study is very well presented and the results summarised accurately and concisely.

The authors have also been careful in their analysis to account for allelic mapping bias in the sQTL and allele specific expression (ASE) analyses, by generating additional alignments using WASP for each tissue, that remove any variants showing significant reference mapping bias.

The authors have also made all the data from the study publicly available, including summary statistics from GWAS. This sets an excellent and encouraging precedent for studies of this type in farmed animals.

I just have a few specific line changes for the authors to address:

Line 42 please add detail as to the meaning of 'undisturbed' mammalian fertilisation, the term itself is not very self explanatory.

Line 58-59 Add additional reference that would have been published after this manuscript was submitted [https://www.cell.com/cell-genomics/pdf/S2666-979X\(23\)00182-9.pdf](https://www.cell.com/cell-genomics/pdf/S2666-979X(23)00182-9.pdf). It would be useful to make specific mention of the findings from this work related to splicing QTLs.

In figure 1D clustering of the expression levels for the three different tissue types are shown. Ideally in the supplemental materials the three tissue types should be plotted separately to identify any outlying samples within each group, and any batch effects associated with the time taken to snap freeze the tissues. This is important because the lowest RIN values were <4 and this level of degradation could have a significant effect on gene expression, and introduce some level (though likely minimal) level of noise into the results.

Line 113 the full stop is on the wrong side of the bracket after 'E'.

Line 127 The relatively low expression of NANOS2 in testes is mentioned but not the high expression of DAZL in the testis tissue, and the tissue specific expression pattern, both genes are relevant in the creation of germ line ablated surrogate sires and while not surprising the high expression of DAZL and it's tissue-specificity in post pubertal bulls would be helpful to mention.

Line 130 Could the authors include the original reference for this dataset if one exists as well as the cattle GTEx reference.

Line 134 How many samples from the cattle GTEx dataset were from post-pubertal (1.5 – 9 years) Bos taurus taurus bulls.

Line 137-138 Other factors could contribute to these differences too including collection and storage of the tissues, and quality of the RNA.

Line 146 Change 'was' to 'were'

Line 471 Change 'was' to 'has been'

Line 531-532 Was there any batch effect observed between the samples collected within 40 minutes and those collected at 270 minutes? This is useful information for transcriptomic studies in farmed animals where collecting tissue samples immediately isn't always feasible.

Line 546 Was the additional 60 minute waiting period/incubation on ice? Or is this a typo and should this be a 60 second waiting period?

Line 549 A RIN value of <4 is very low and the RNA likely to be degraded. Could the authors include which samples had the low RIN values in the supplementary materials, did these low values correlate with the samples where time to flash freezing after euthanasia was longer or a specific tissue. Did the authors perform a PCA or any cluster analysis of the TPMs of the RNA-Seq samples based on RIN values for each group to identify any potential batch effects?

Line 673 I can see RIN was included as co-variate in the sQTL analysis. How different were the RIN values across samples? Could this be included in additional supplementary file? Apologies if this already exists and I missed it.

Line 754-755 Add 'in this study for testes'. Unless Kallisto refers to RefSeq annotation transcript quantification?

Reviewer #2 (Remarks to the Author):

The manuscript is generally well-written and relatively easy to follow if you are familiar with the theme. The authors generated an impressive dataset (number of samples and depth of sequencing), and I can imagine that this submission is only part of the studies that will derive from it.

There is so much data (and analyses) that the manuscript became a descriptive list of the different analyses/results. The rationale and order of the presentation make sense, but I felt that it missed a "punch" of achievement/contribution to the field. Then, towards the end, I noticed that all data was made publicly – I believe you could explore this fact better in the manuscript; from the abstract to the beginning of the results, I think you should highlight that this data was generated and is available to the scientific community.

What is the key message of the article? What was its objective?

Is it a descriptive article or applied male fertility? I don't think there is a right or wrong answer here. The title hits a more applied work, but most of the results and discussion are on comparative and descriptive work.

My only hard criticism is about the choice not to include chromosome X in any analyses. It is known that this chromosome has several QTL for male and female fertility traits. Genome predictions for cattle fertility traits are also known to be influenced by the inclusion (or not) of markers in BTAX. A justification should be provided for not including the X chromosome in the analyses. I know it might be "painful" and require taking quite a few steps back, but I think it would be good to pause and discuss this point.

I strongly suggest reducing the number of abbreviations. If you use the abbreviation only once or twice in the whole document, you don't need an abbreviation.

Specific comments.

Title. I understand the attempt to make cattle-based research more "palatable" to the general scientific community, but I am unsure if I agree with the approach for calling cattle or livestock a "large mammal".

Abstract line 2 and throughout the text. "artificial insemination bull". I know what you are talking about, but scientists outside animal science will not. Maybe use a different sentence in the abstract, then define the term in the introduction.

Abstract and throughout the text. "molecular phenotype". What are them? This need to be better explained.

Line 54. Maybe missing a word in this sentence after "molecular"?

Line 66. In addition to uniform cohorts, you could add that also required a relatively large sample size.

Line 82. Could potentially add the citation for the cattle genome you used.

Line 100, also in Methods. (≥ 0.1 TPM and ≥ 6 reads in $\geq 20\%$ of samples). Why in $\geq 20\%$ of samples? The 20% will vary depending on the tissue because there is a variable number of samples in each tissue; you are saying that testis needs ~ 23 animals expressing that transcript to be considered "real", but for vas deferens, only ~ 17 . Wouldn't be more consistent just to apply a threshold across all tissues?

Figures. I liked the consistency of tissue colour across the figures. But I should note that the pale blue and pale purple sometimes tricked my eyes. I should not, though, the figures are well-worked and look good.

Table 1. Maybe this is not the best location for this table.

Table 1. All of those abbreviations might need to be defined!

Line 119. Supporting File 1. The file I had access, had only results related to one tissue (epididymis-specificity index). For completion, the file should include also the tissue specific or enriched genes for the other tissues.

Line 136. Expression differences were likely due to different RNA sequencing strategies. How do you support this statement? Why only the differences were influenced by the technology used? The fact that the testis is very complex, with different specified tissue layers and dynamic, and the sampling strategy might play a major role...

Line 166. Adding more information. 11,087 genes had alternative splicings. How many isoforms were found? 2 to ?? What was the gene with the largest number of isoforms? Was it also expressed in other tissues? Is it also highly spliced there?

Line 187. Only autosomal variants were explored, right?

Line 194. The fact that more eGenes were found in testis compared to the other tissues; could be due to the larger sample size?

Line 211. What is the definition of non-coding transcript? Often, something upstream or downstream is also considered non-coding. Possibly picking at random 172 transcripts from upstream or downstream, you might get a large and similar effect to those non-coding.

Line 216. sQTL; eQTL. Might need to define.

Line 225. sVariants or eVariants?

Line 286. molQTL might need to define.

Line 288. No reference to BovReg? No publication, no repro tissue? Or this is a resource from BovReg?

Line 330. Only autosomal SNP. Should be noted.

Line 330. First time direct "effect" on male fertility – should explain the male fertility trait here.

Line 341. 342. MetaXcan, S-PrediXcan. Is there a citation for them? Should be included here as well.

Line 447 – 451. Carefully consider using the same word when talking about humans and cattle.

does the heterogeneity of cattle (diff cattle breeds) correspond to the heterogeneity in humans (diff populations)?

Line 527. The timing between sample collection and processing (freezing) was higher than expected for an RNA sequencing analysis. Did you test the effect of the time length between killing and freezing on RNA quality?

Coincidentally, vas deferens were the tissue with a "smaller gene expression" but also lower RIN values for the quality of RNA (average RIN=4, Which is quite low). Did you test the effect of the RIN values on gene expression? I can see that for some analyses, the RIN values were considered but maybe it was not considered in all of them? How to remove the potentially confounding effect of lower RIN in the lower gene expression?

Was there an effect of the season of the year on the gene expression? We know that reproductive are potentially less "active" in winter compared to spring.

Reviewer #3 (Remarks to the Author):

Response to Reviewer Comments

Reviewer #1 (Remarks to the Author):

This study uses whole genome sequencing and transcriptomic analyses of testis, epididymis, and vas deferens tissues to perform eQTL, splice eQTL and TWAS analyses to understand the molecular drivers of fertility traits in bulls of Braunvieh ancestry.

There are very few datasets with matched whole genome sequence and transcriptomic data of this scale and quality for farmed animals and the authors emphasise the opportunity that the study provides in leveraging this type of data for complex trait prediction, particularly for a trait which is mid- to -lowly heritably such as fertility.

The study is very well presented and the results summarised accurately and concisely.

The authors have also been careful in their analysis to account for allelic mapping bias in the sQTL and allele specific expression (ASE) analyses, by generating additional alignments using WASP for each tissue, that remove any variants showing significant reference mapping bias.

The authors have also made all the data from the study publicly available, including summary statistics from GWAS. This sets an excellent and encouraging precedent for studies of this type in farmed animals.

AUTHOR: We thank the reviewer for their encouraging assessment of our study! We've addressed all comments below and implemented the necessary changes in the manuscript.

I just have a few specific line changes for the authors to address:

Line 42 please add detail as to the meaning of 'undisturbed' mammalian fertilisation, the term itself is not very self explanatory.

AUTHOR: We agree. We have deleted the word "undisturbed", which is not needed in this context.

Line 58-59 Add additional reference that would have been published after this manuscript was submitted [https://www.cell.com/cell-genomics/pdf/S2666-979X\(23\)00182-9.pdf](https://www.cell.com/cell-genomics/pdf/S2666-979X(23)00182-9.pdf). It would be useful to make specific mention of the findings from this work related to splicing QTLs.

AUTHOR: We have added this reference to lines 56–57.

In figure 1D clustering of the expression levels for the three different tissue types are shown. Ideally in the supplemental materials the three tissue types should be plotted separately to identify any outlying samples within each group, and any batch effects associated with the time taken to snap freeze the tissues. This is important because the lowest RIN values were <4 and this level of degradation could have a significant effect on gene expression, and introduce some level (though likely minimal) level of noise into the results.

AUTHOR: We thank the reviewer for their attention to possible batch effects, which are an important aspect to consider in gene expression and splicing analyses. RIN values were indeed lower for vas deferens than the other tissues. However, we considered RIN (as well as PEER factors, age, and principal components from a genomic relationship matrix) as covariates in our e/sQTL analyses to account for batch effects and possible hidden confounders; thus, we are confident that our results are not affected by batch effects. We have included PCAs of normalized expression and splicing values for each individual tissue type in Supplemental Figure 4C (expression) and Supplemental Figure 4D (splicing) of the original submission. We now refer to these figures in lines 606–607 for expression and lines 624–625 for splicing, and mention that we observed no obvious batch effects or outliers. Also, we now provide exhaustive metadata in an Excel spread sheet as a new supplemental table (Table S1). To further test if samples clustered according to confounding factors (specifically RIN values, time between euthanasia and flash freezing, and month sampled (as a proxy for season)), we performed clustering analyses on the raw expression phenotypes within each individual tissue. We observed no apparent clustering according to RIN value, particularly for

vas deferens. Epididymis contained three clusters that were separated by the first and the second principal components. We suspect this clustering points to regional-specific gene expression that occurs in the head of the epididymis^{1,2}. Though this pattern was evident when plotting the epididymis samples alone, overall expression between these clusters was more similar than expression in the other two tissues (Figure 1D). We show the raw expression PCAs with RIN value, time between euthanasia and freezing (in minutes), and month of sample for each individual tissue in Figure R1.

Figure R1: Association between the top principal components from the raw TPM values and RIN, time between slaughter and sampling, and sampling season for the three tissues.

Line 113 the full stop is on the wrong side of the bracket after 'E'.

AUTHOR: Fixed.

Line 127 The relatively low expression of NANOS2 in testes is mentioned but not the high expression of DAZL in the testis tissue, and the tissue specific expression pattern, both genes are relevant in the

creation of germ line ablated surrogate sires and while not surprising the high expression of *DAZL* and it's tissue-specificity in post pubertal bulls would be helpful to mention.

AUTHOR: The paragraph the reviewer is referring to investigates differences in gene expression in the male reproductive tract of humans, mice, and cattle. Our analyses showed that *NANOS2* is not expressed in our testis cohort, whereas it is moderately expressed in humans. We confirm that *DAZL* is highly expressed in our bovine testis cohort (\emptyset TPM: 163), however, it is also highly expressed in human testis (\emptyset TPM: 223). Since our analysis did not reveal differential expression of *DAZL* between species, we don't think that there's a reason to mention *DAZL* expression in the main manuscript text. In any case, the TPM matrices (both raw and normalized) are available at the zenodo repository (<https://zenodo.org/records/10053815>).

Line 130 Could the authors include the original reference for this dataset if one exists as well as the cattle GTEX reference.

AUTHOR: There is not really an original reference for this dataset, as it is a subset of the data processed by cattle GTEX. We have included a reference to cattle GTEX already in the original manuscript. Cattle GTEX aggregated existing data from public repositories (please note that none of the authors of this manuscript are members of the farm GTEX consortium). Supplementary Table 4 from cattle GTEX reports that 60 testis samples fulfilled their criteria for downstream analyses, and this is the number we mention here. Supplementary Table 1 in cattle GTEX lists all data sources that were considered for their analyses. For the sake of completeness, **Table R1** lists the IDs of the samples we considered for our comparison (these are the IDs provided in Supplementary Table 1 of cattle GTEX).

Table R1: Sequence read archive identifiers of 10 cattle GTEX testis transcriptomes from post-pubertal bulls.

sample	Bioproject	breed	age
SRS3309609	PRJNA471564	Angus	1.5 years
SRS3309607	PRJNA471564	Angus	1.5 years
SRS3309608	PRJNA471564	Angus	1.5 years
SRS3021383	PRJNA437027	Chinese yellow cattle	4 years
CRS013209	PRJCA000530	Chinese yellow cattle	4~6y
CRS013202	PRJCA000530	Chinese yellow cattle	4~6y
CRS013216	PRJCA000530	Chinese yellow cattle	4~6y
CRS013223	PRJCA000530	Chinese yellow cattle	4~6y
SRS734278	PRJNA386670	Hereford	9 years
SRS485384	NA	Unknown	2 years

Line 134 How many samples from the cattle GTEX dataset were from post-pubertal (1.5 – 9 years) *Bos taurus taurus* bulls.

AUTHOR: Ten – see **line 135**.

Line 137-138 Other factors could contribute to these differences too including collection and storage of the tissues, and quality of the RNA.

AUTHOR: We agree that these factors could also contribute to the differences in gene expression between our dataset and Cattle GTEX. They have been added to **lines 138–141**.

Line 146 Change 'was' to 'were'

AUTHOR: Fixed.

Line 471 Change 'was' to 'has been'

AUTHOR: Fixed.

Line 531-532 Was there any batch effect observed between the samples collected within 40 minutes and those collected at 270 minutes? This is useful information for transcriptomic studies in farmed animals where collecting tissue samples immediately isn't always feasible.

AUTHOR: We did not observe any apparent batch effects associated with the length of time between sample collection and freezing. To support this, we conducted PCAs with the raw expression values and observed no clustering related to sample collection time (see **Figure R1** presented above). In addition, we now provide a table of each sample's metadata, which includes the length of time between collection and freezing (**Table S1**).

Line 546 Was the additional 60 minute waiting period/incubation on ice? Or is this a typo and should this be a 60 second waiting period?

AUTHOR: The waiting period was 60 minutes to ensure complete lysing. The samples were kept in a benchtop cold block during the waiting period to reduce degradation. This detail has been added to **line 553**.

Line 549 A RIN value of <4 is very low and the RNA likely to be degraded. Could the authors include which samples had the low RIN values in the supplementary materials, did these low values correlate with the samples where time to flash freezing after euthanasia was longer or a specific tissue. Did the authors perform a PCA or any cluster analysis of the TPMs of the RNA-Seq samples based on RIN values for each group to identify any potential batch effects?

AUTHOR: We have included the RIN values for each sample in the meta data table, which is now provided in the supplement (**Table S1**). We observed little-to-no correlation between sampling time and RIN value in all three tissues (Spearman's rho: testis = 0.11; epididymis = -0.14; vas deferens = -0.02). Vas deferens had lower RIN values than testis and epididymis (**Table S3**). Prior to our eQTL and sQTL discovery, we performed clustering analyses with a variety of covariates and did not identify any obvious batch effects (see **Figure R1**). We considered RIN, PEER factors, age, and principal components from a genomic relationship matrix as covariates in our e/sQTL analyses to account for batch effects and possible hidden confounders; thus, we are confident that our results are not affected by batch effects.

Line 673 I can see RIN was included as co-variate in the sQTL analysis. How different were the RIN values across samples? Could this be included in additional supplementary file? Apologies if this already exists and I missed it.

AUTHOR: We have now included a new supplementary table with exhaustive meta data including the RIN value for each RNA sample (**Table S1**). To visualize the distribution of RIN values for each tissue type, we plotted histograms (**Figure R2**). The black line represents the median RIN value (Testis = 9.0 ± 0.5 ; Epididymis = 8.4 ± 0.9 ; Vas deferens = 5.9 ± 1.1 ; Table S3). RIN was a covariate in all our e/sQTL analyses, which accounts for the vast majority of possible degradation effects³.

Figure R2: Distribution of RIN values for the three tissues.

Line 754-755 Add 'in this study for testes'. Unless Kallisto refers to RefSeq annotation transcript quantification?

AUTHOR: Done.

Reviewer #2 (Remarks to the Author):

The manuscript is generally well-written and relatively easy to follow if you are familiar with the theme. The authors generated an impressive dataset (number of samples and depth of sequencing), and I can imagine that this submission is only part of the studies that will derive from it.

AUTHOR: We thank the reviewer for their supportive and constructive assessment of our study!

There is so much data (and analyses) that the manuscript became a descriptive list of the different analyses/results. The rationale and order of the presentation make sense, but I felt that it missed a "punch" of achievement/contribution to the field. Then, towards the end, I noticed that all data was made publicly – I believe you could explore this fact better in the manuscript; from the abstract to the beginning of the results, I think you should highlight that this data was generated and is available to the scientific community.

What is the key message of the article? What was its objective?

Is it a descriptive article or applied male fertility? I don't think there is a right or wrong answer here. The title hits a more applied work, but most of the results and discussion are on comparative and descriptive work.

AUTHOR: It is true that a huge amount of data underpins the results reported in this manuscript, including whole-genome DNA and RNA sequencing data for the entire cohort, TPM/splicing matrices, and GWAS/TWAS summary statistics. These data are publicly available at zenodo and the ENA, which is detailed in the Data Availability section. The release of raw data is mandatory for papers published by Nature Communications; thus, we believe our compliance with this prerequisite does not require more attention in the manuscript.

The key messages of our article are:

- Homogeneous cohorts enable a much more comprehensive transcriptomic profiling than heterogeneous cohorts. We think this finding is highly relevant for the ever-increasing number of GTEX-like studies that rely on highly heterogeneous cohorts. This message is prominently placed in the first paragraph of the discussion section.
- Our cohort enables us to study differences and similarities in three male reproductive tissues in a mammalian species, which was barely possible so far due to the lack of epididymis and vas deferens samples. We also confirm the transcriptional complexity of testis compared to the other tissues. We've added a new analysis based on the suggestion of this reviewer that considers 74 bulls that have RNA samples for all tissues to support this finding.
- By integrating GWAS and e/sQTL data through TWAS and colocalization, we were able to help prioritize the numerous causal variants and molecular phenotypes underpinning male fertility QTL.

We carefully went through the manuscript and made several changes in the introduction to help clarify the objectives of the work.

My only hard criticism is about the choice not to include chromosome X in any analyses. It is known that this chromosome has several QTL for male and female fertility traits. Genome predictions for cattle fertility traits are also known to be influenced by the inclusion (or not) of markers in BTAX. A justification should be provided for not including the X chromosome in the analyses. I know it might be “painful” and require taking quite a few steps back, but I think it would be good to pause and discuss this point.

AUTHOR: We thank the reviewer for their comment on the missing X chromosomal analysis. However, our gene expression matrix and analyses included X chromosomal genes. Our comparative analysis based off human and murine data published by Robertson et al.⁴ and Djureinovic et al.⁵ also includes X chromosomal genes. We agree that this was not evident from the description in the original manuscript. This has now been clarified in the main text and **Table 1**. However, we noticed that our splicing matrix and clustering analysis did not contain the X chromosome, as it is removed by default with the 'prepare_phenotype_table.py' script from Leafcutter which we applied to compile splicing variation across genes and samples. We modified the 'prepare_phenotype_table.py' script accordingly and now include the X chromosome in the splicing matrix, splicing analyses, and clustering analyses, and have updated the corresponding figures and tables (**Figure 1D/E**, **Table 1**, **Figure S4**), and the corresponding text in the main manuscript (**lines 168–178**). We have also updated the zenodo repository accordingly; the current version of the repository (<https://zenodo.org/records/10053815>) contains splicing files that include the X chromosome.

Figure R3 shows that the number of expressed genes is similar for the X chromosome and autosomes of similar length. We observed no notable differences in gene expression and splicing patterns between the autosomes and the X chromosome. Both chromosome sets showed the same general gene expression and splicing patterns; for example, in both sets, epididymis had the most expressed genes, testis had the most spliced genes, and vas deferens had the fewest expressed and spliced genes (**Table R2**). These numbers are also mentioned in the revised manuscript both in the main text (**lines 101–102**, **lines 168–178**) and in **Table 1**.

Figure R3: Correlation between chromosome length and number of expressed genes for three reproductive tissues.

Table R2: Number of expressed and spliced genes on autosomes and the X chromosome.

	Testis	Epididymis	Vas deferens
Expressed Genes (Autosomal Chromosomes)	19440	19561	18328
Expressed Genes	747	782	705

(X Chromosome)			
Spliced Genes (Autosomal Chromosomes)	14243	13558	12332
Spliced Genes (X Chromosome)	481	468	448

A recent study conducted by our group showed that many bovine X chromosomal variants are missed by short reads⁶; therefore, we believe X- (as well as Y-) chromosomal e/sQTL can not be readily compared to those identified along the autosomes. Advances in sequencing and assembly technologies enable complete assembly of sex chromosomes⁷, and near-error-free bovine sex chromosome assemblies are currently being produced and annotated (e.g., GCF_002263795.3_ARS-UCD2.0). Accurate and complete annotations, together with long read sequencing data, are required to thoroughly characterize e/sQTL on the sex chromosomes. This analysis is not feasible with the data we present in our manuscript, and thus, is beyond the scope of the current work.

Furthermore, we have previously conducted a GWAS between male fertility traits and X-chromosomal sequence variants (from genotype arrays) in a large cohort of Brown Swiss bulls. This analysis did not reveal X chromosomal QTL (see Hiltpold et al., <https://www.research-collection.ethz.ch/handle/20.500.11850/549843>, pp166-171). While we acknowledge that X-chromosomal variants may have large effects on male fertility in certain breeds of cattle (e.g., Fortes et al.⁸), this appears to not be the case for the Braunvieh breed studied here.

While we think an X chromosomal e/sQTL analyses currently has substantial limitations for the reasons mentioned above—and therefore didn't include in the current manuscript—the gene expression and splicing matrices we deposited at zenodo contain TPM and PSI values for X chromosomal genes.

I strongly suggest reducing the number of abbreviations. If you use the abbreviation only once or twice in the whole document, you don't need an abbreviation.

AUTHOR: We carefully went over the manuscript and removed unnecessary abbreviations. This is reflected by the removal of nine barely used abbreviations (WGS, BV, OB, LD, ICSS, GO, TWAS, LFSR, VEP). The revised manuscript contains only widely used abbreviations that are mentioned several times (GWAS, QTL, GTEx, molQTL, eQTL, sQTL, DNA, RNA, PCA, RIN, TPM, FDR), and we carefully checked that all are introduced upon first appearance.

Specific comments.

Title. I understand the attempt to make cattle-based research more “palatable” to the general scientific community, but I am unsure if I agree with the approach for calling cattle or livestock a “large mammal”.

AUTHOR: Changed to “cattle”.

Abstract line 2 and throughout the text. “artificial insemination bull”. I know what you are talking about, but scientists outside animal science will not. Maybe use a different sentence in the abstract, then define the term in the introduction.

AUTHOR: We've changed the beginning of the abstract to “Breeding bulls are well suited to investigate inherited variation in male fertility because they are genotyped and their reproductive success is monitored through semen analyses and thousands of artificial inseminations.” We also define artificial insemination bulls in **lines 38–39**.

Abstract and throughout the text. “molecular phenotype”. What are them? This need to be better explained.

AUTHOR: We thank the reviewer for their comment. We rephrased the paragraph where we introduce “molecular phenotypes” for the first time to clarify that molecular phenotypes are derived from transcriptomic data (**lines 55–59**). However, we think that “molecular phenotype” is a widely used term that doesn’t need too much further explanation.

Line 54. Maybe missing a word in this sentence after “molecular”?

AUTHOR: Correct as is: «... molecular underpinnings ...». (**line 52**)

Line 66. In addition to uniform cohorts, you could add that also required a relatively large sample size.

AUTHOR: Agreed – we’ve added «large». (**line 62**)

Line 82. Could potentially add the citation for the cattle genome you used.

AUTHOR: We have added the abbreviation to the reference (ARS-UCD1.2; **line 82**) but provide the reference to Rosen et al., in the Material and Methods section (**line 568**).

Line 100, also in Methods. (≥ 0.1 TPM and ≥ 6 reads in $\geq 20\%$ of samples). Why in $\geq 20\%$ of samples? The 20% will vary depending on the tissue because there is a variable number of samples in each tissue; you are saying that testis needs ~23 animals expressing that transcript to be considered “real”, but for vas deferens, only ~17. Wouldn’t be more consistent just to apply a threshold across all tissues?

AUTHOR: We preferred to apply a consistent %-threshold across all samples. Thus, we opted for the 20% rather than different numbers across the three tissues. This method has been applied to other large cohorts that analyzed tissues with different sample sizes, specifically human GTEx⁹.

Figures. I liked the consistency of tissue colour across the figures. But I should note that the pale blue and pale purple sometimes tricked my eyes. I should not, though, the figures are well-worked and look good.

AUTHOR: We are happy to see the reviewer appreciates the consistent color scheme we applied to facilitate interpretation of our results!

Table 1. Maybe this is not the best location for this table.

AUTHOR: We suppose that the table will be positioned at an appropriate place during the final copy-editing step. For now, we have moved the table to **line 264**, which is after the introduction of sQTL.

Table 1. All of those abbreviations might need to be defined!

AUTHOR: All abbreviations have been introduced upon first appearance in the main text and before Table 1 in the revised manuscript (eQTL: **line 193**; eGene: **line 200**; eVariant: **line 210**, sQTL; **line 247**; sGene: **line 250**; sVariant: **line 257**). Figure 2 was also moved to ensure all abbreviations are defined prior to its appearance (**line 277**).

Line 119. Supporting File 1. The file I had access, had only results related to one tissue (epididymis-specificity index). For completion, the file should include also the tissue specific or enriched genes for the other tissues.

AUTHOR: The requested information was included in the original submission. The source file (excel spread sheet) we uploaded contains three separate sheets for the three tissues (**Figure R4**).

33	ENSBTAG00000048650	129.8	0	0	Inf	ENSBTAG00000048650.1		
34	ENSBTAG00000051173	120.8	0	0.8	149.9	ENSBTAG00000051173.1		
35	ENSBTAG00000021913	95.4	0.8	0.7	113	MMD2		
36	ENSBTAG00000033166	93.8	1.1	0.8	84.8	ENSBTAG00000033166.4		
37	ENSBTAG00000048009	87.6	0	0	Inf	ENSBTAG00000048009.2		
38	ENSBTAG00000043583	84.6	0	0	Inf	ENSBTAG00000043583.1		
39	ENSBTAG00000030542	65	0.2	0	322.5	KRT25		

Testis-enriched expression **Epididymis-enriched expression** Vas deferens-enriched expression +

Figure R4: Screenshot showing the three different sheets included in Supporting File 1.

Line 136. Expression differences were likely due to different RNA sequencing strategies. How do you support this statement? Why only the differences were influenced by the technology used? The fact that the testis is very complex, with different specified tissue layers and dynamic, and the sampling strategy might play a major role...

AUTHOR: While we think that the stranded RNA sequencing strategy applied to our cohort enabled us to more accurately analyze gene expression and splicing, we agree that other factors could have also contributed to the differences in gene expression between our dataset and cattle GTEx. This has been clarified in **lines 138–141**.

Line 166. Adding more information. 11,087 genes had alternative splicings. How many isoforms were found? 2 to ?? What was the gene with the largest number of isoforms? Was it also expressed in other tissues? Is it also highly spliced there?

AUTHOR: We report results on the gene-level based on the current annotation of the bovine genome throughout the paper. Leafcutter infers variable splicing events without relying on known isoform annotations¹⁰. However, Leafcutter does not perform a de novo annotation, and thus, it does not infer novel isoforms. To avoid confusion, we rephrased the text in the main manuscript and now refer to «RNA splicing variation» rather than «alternative splicing» throughout (e.g., **lines 168-172**). **Figure 1E** shows the overlap of spliced genes across the three tissues.

Line 187. Only autosomal variants were explored, right?

AUTHOR: Correct. This has been clarified in the revised manuscript (**lines 197–199**).

Line 194. The fact that more eGenes were found in testis compared to the other tissues; could be due to the larger sample size?

AUTHOR: This is a very valuable comment! To further support that the difference in sample size is not driving the increased number of eGenes in testis, we subset our dataset to include only the 74 samples that had expression data in all three tissues. We observed that testis still had more eGenes than epididymis and vas deferens, despite all three tissues having the same sample size (Table R3). We now mention this in **lines 204–208**. Our findings indicate that testis has very high transcriptional complexity, which confirms results from human GTEx⁹.

Table R3: eGenes identified in the full eQTL analysis, which contained all samples that passed our quality criteria for each tissue, and eGenes identified in a subset of 74 animals with expression data in every tissue.

	Testis	Epididymis	Vas deferens
eGenes (full sampling)	11,164	4,347	3,889
eGenes (74 sample subset)	6,646	3,473	2,041

Line 211. What is the definition of non-coding transcript? Often, something upstream or downstream is also considered non-coding. Possibly picking at random 172 transcripts from upstream or downstream, you might get a large and similar effect to those non-coding.

AUTHOR: Non-coding transcript variants include "non-coding transcript exon variants" and "non-coding transcript variants". Each is specifically defined by Ensembl VEP as: non-coding transcript exon variant, a sequence variant that changes non-coding exon sequence in a non-coding transcript; non-

coding transcript variant, a transcript variant of a non-coding RNA gene. Thus, these variants lie within the exons or transcripts of non-coding RNA and are not considered upstream or downstream by VEP. We have clarified this in **lines 224–225**.

Line 216. sQTL; eQTL. Might need to define.

AUTHOR: s/eQTL are defined in lines 193 and 247.

Line 225. sVariants or eVariants?

AUTHOR: sVariants is correct

Line 286. molQTL might need to define.

AUTHOR: The abbreviation is defined in the introduction in the revised manuscript (line 55).

Line 288. No reference to BovReg? No publication, no repro tissue? Or this is a resource from BovReg?

AUTHOR: The data we present in our manuscript were collected independently from Bovreg. To the best of our knowledge, there is no publication available yet from Bovreg that supports this statement.

Line 330. Only autosomal SNP. Should be noted.

AUTHOR: Clarified at several places throughout the manuscript.

Line 330. First time direct “effect” on male fertility – should explain the male fertility trait here.

AUTHOR: The trait is explained in the subsequent sentence: “The fertility of the bulls was quantified through the proportion of successful artificial inseminations [...]” (line 344)

Line 341. 342. MetaXcan, S-PrediXcan. Is there a citation for them? Should be included here as well.

AUTHOR: We have added the citation for MetaXcan to lines 353. This is the same citation for S-PrediXcan, which is specifically mentioned in the Methods (lines 727).

Line 447 – 451. Carefully consider using the same word when talking about humans and cattle. does the heterogeneity of cattle (diff cattle breeds) correspond to the heterogeneity in humans (diff populations)?

AUTHOR: Agreed. We now write the “heterogeneity of cattle GTEx which contains individuals from different populations sampled at different ages...” and “...similar to what was reported in a three-times larger—human GTEx cohort that also contained individuals with diverse ancestries sampled at different ages,” (lines 460–461 and 464–465).

Line 527. The timing between sample collection and processing (freezing) was higher than expected for an RNA sequencing analysis. Did you test the effect of the time length between killing and freezing on RNA quality?

Coincidentally, vas deferens were the tissue with a “smaller gene expression” but also lower RIN values for the quality of RNA (average RIN=4, Which is quite low). Did you test the effect of the RIN values on gene expression? I can see that for some analyses, the RIN values were considered but maybe it was not considered in all of them? How to remove the potentially confounding effect of lower RIN in the lower gene expression?

AUTHOR: Samples were collected at a commercial slaughterhouse and subsequently transported to our lab. We did not observe a relationship between processing time and RIN values in any tissue (Spearman’s rho: testis = 0.11; epididymis = -0.14; vas deferens = -0.02). RIN values were indeed lower for vas deferens than the other tissues. We have now included a supplementary table with the RIN value for each sample (Table S1). To visualize the distribution of RIN values for each tissue type,

we plotted histograms (**Figure R2**). The black line represents the median RIN value (Testis = 9.0 ± 0.5 ; Epididymis = 8.4 ± 0.9 ; Vas deferens = 5.9 ± 1.1 ; Table S3).

Figure R2 (copied from above): Distribution of RIN values for the three tissues.

To further test if samples clustered according to confounding factors, specifically RIN values, time between euthanasia and flash freezing, and month of sampling (to account for season), we also performed clustering analyses on the raw expression phenotypes within each individual tissue (**Figure R1**). We observed no clear clustering according to RIN value, particularly for vas deferens. Epididymis contained three clusters that were separated by the first and the second principal components. We suspect this clustering points to regional-specific gene expression that occurs in the head of the epididymis^{1,2}. Though this pattern is evident when plotting the epididymis samples individually, overall expression between these clusters is more similar than expression in the other two tissues (**Figure 1D**).

RIN was a covariate in all our e/sQTL models, which accounts for the vast majority of possible degradation effects³. RIN was also accounted for in subsequent analyses, specifically in our estimates of effect size (as a covariate in aFC) and TWAS (through the summary statistics of the e/sQTL analyses).

Figure R1 (copied from above): Association between the top principal components obtained from TPM values and RIN, time between slaughter and sampling, and month of sampling for the three tissues.

Was there an effect of the season of the year on the gene expression? We know that reproductive are potentially less “active” in winter compared to spring.

AUTHOR: This was not tested in our study, though we did not observe any clustering according to sampling month in our PCAs (see **Figure R1** presented above). We now provide a spread sheet with exhaustive meta data including sampling date as **Table S1**, so possible impacts of sampling season on gene expression may be investigated in follow-up studies.

Reviewer #3 (Remarks to the Author):

AUTHOR: Thank you for your contribution to the review of our manuscript!

References

1. Guyonnet, B. *et al.* The adult boar testicular and epididymal transcriptomes. *BMC Genomics* **10**, (2009).
2. Johnston, D. S. *et al.* The mouse epididymal transcriptome: Transcriptional profiling of segmental gene expression in the epididymis. *Biol Reprod* **73**, 404–413 (2005).
3. Gallego Romero, I., Pai, A. A., Tung, J. & Gilad, Y. RNA-seq: Impact of RNA degradation on transcript quantification. *BMC Biol* **12**, (2014).
4. Robertson, M. J. *et al.* Large-scale discovery of male reproductive tract-specific genes through analysis of RNA-seq datasets. *BMC Biol* **18**, (2020).
5. Djureinovic, D. *et al.* The human testis-specific proteome defined by transcriptomics and antibody-based profiling. *Mol Hum Reprod* **20**, 476–488 (2014).
6. Leonard, A. S., Mapel, X. M. & Pausch, H. Pangenome genotyped structural variation improves molecular phenotype mapping in cattle. *bioRxiv* (2023) doi:10.1101/2023.06.21.545879.
7. Rhie, A. *et al.* The complete sequence of a human Y chromosome. *Nature* **621**, 344–354 (2023).
8. Fortes, M. R. S. *et al.* X chromosome variants are associated with male fertility traits in two bovine populations. *Genetics Selection Evolution* **52**, (2020).
9. GTEx Consortium. The GTEx Consortium atlas of genetic regulatory effects across human tissues. *Science (1979)* **369**, 1318–1330 (2020).
10. Li, Y. I. *et al.* Annotation-free quantification of RNA splicing using LeafCutter. *Nat Genet* **50**, 151–158 (2018).

Reviewer #1 (Remarks to the Author):

The authors have comprehensively addressed all of the comments I raised and have provided a clear rebuttal and revised manuscript. I am happy that the manuscript can now proceed for publication.

Reviewer #2 (Remarks to the Author):

The authors did a comprehensive review of their submission and improved this revised version in light of the reviewers' comments. All of the reviewers' comments were considered, discussed in the reply letter, and, when appropriated, incorporated into the text. I have no further comments.

Reviewer #3 (Remarks to the Author):

I appreciate the authors efforts in reviewing the manuscript. The questions and comments were addressed properly and comprehensively and the additional Analysis run in response to the reviewers's comments certainly added credibility to the manuscript.

Point-by-point response to the reviewers' comments

Reviewer #1 (Remarks to the Author):

The authors have comprehensively addressed all of the comments I raised and have provided a clear rebuttal and revised manuscript. I am happy that the manuscript can now proceed for publication.

Authors: Thank you for your contribution to the review process and your valuable suggestions that helped improving our manuscript!

Reviewer #2 (Remarks to the Author):

The authors did a comprehensive review of their submission and improved this revised version in light of the reviewers' comments. All of the reviewers' comments were considered, discussed in the reply letter, and, when appropriated, incorporated into the text. I have no further comments.

Authors: Thank you for your contribution to the review process and your valuable suggestions that helped improving our manuscript!

Reviewer #3 (Remarks to the Author):

I appreciate the authors efforts in reviewing the manuscript. The questions and comments were addressed properly and comprehensively and the additional Analysis run in response to the reviewers's comments certainly added credibility to the manuscript.

Authors: Thank you for your contribution to the review process and your valuable suggestions that helped improving our manuscript!